# In vitro proteasome processing of neo-splicetopes does not predict their presentation in vivo

Gerald Willimsky[1,2,3]*, Christin Beier[4], Lena Immisch[1,2,3,5], George Papafotiou[1,2,3], Vivian Scheuplein[6], Andrean Goede[7], Hermann-Georg Holzhütter[4], Thomas Blankenstein[1,6,8†], Peter M Kloetzel[4†]*

[1]Institute of Immunology, Charité-Universitätsmedizin Berlin, corporate member of Freie Universität Berlin and Humboldt-Universität zu Berlin, Berlin, Germany; [2]German Cancer Research Center, Heidelberg, Germany; [3]German Cancer Consortium, partner site Berlin, Berlin, Germany; [4]Institute of Biochemistry, Charité – Universitätsmedizin Berlin, corporate member of Freie Universität Berlin and Humboldt-Universität zu Berlin, Berlin, Germany; [5]Humboldt-Universität zu Berlin, Berlin, Germany; [6]Max Delbrück Center for Molecular Medicine in Helmholtz Association, Berlin, Germany; [7]Institute of Physiology, Charité – Universitätsmedizin Berlin, corporate member of Freie Universität Berlin and Humboldt-Universität zu Berlin, Berlin, Germany; [8]Berlin Institute of Health, Berlin, Germany

*For correspondence:
gerald.willimsky@charite.de (GW);
p-m.kloetzel@charite.de (PMK)

†These authors contributed equally to this work

Competing interests: The authors declare that no competing interests exist.

**Abstract** Proteasome-catalyzed peptide splicing (PCPS) of cancer-driving antigens could generate attractive neoepitopes to be targeted by T cell receptor (TCR)-based adoptive T cell therapy. Based on a spliced peptide prediction algorithm, TCRs were generated against putative KRAS[G12V]- and RAC2[P29L]-derived neo-splicetopes with high HLA-A*02:01 binding affinity. TCRs generated in mice with a diverse human TCR repertoire specifically recognized the respective target peptides with high efficacy. However, we failed to detect any neo-splicetope-specific T cell response when testing the in vivo neo-splicetope generation and obtained no experimental evidence that the putative KRAS[G12V]- and RAC2[P29L]-derived neo-splicetopes were naturally processed and presented. Furthermore, only the putative RAC2[P29L]-derived neo-splicetopes was generated by in vitro PCPS. The experiments pose severe questions on the notion that available algorithms or the in vitro PCPS reaction reliably simulate in vivo splicing and argue against the general applicability of an algorithm-driven 'reverse immunology' pipeline for the identification of cancer-specific neo-splicetopes.

## Introduction

Defined anti-tumor CD8[+] T cell responses require the proteasome-dependent processing of intracellular proteins and the efficient generation of antigenic peptides presented in the context of HLA class I molecules at the cell surface for TCR recognition. An important step in defining the proteasome as HLA class I epitope generation machine was the early observation that purified 20S proteasomes in combination with synthetic polypeptide substrates encompassing the epitope of interest reproduced the in vivo generation of these epitopes with high fidelity (*Boes et al., 1994*; *Guillaume et al., 2012*; *Kessler et al., 2006*; *Niedermann et al., 1995*; *van der Bruggen and Van den Eynde, 2006*). Thus, in vitro antigen processing experiments in combination with specific CD8[+] T cells to monitor HLA class I binding and immune recognition are a widely used reliable tool to verify the generation efficiency of antigenic peptides of viral, bacterial, and human origin.

Our view on antigen processing was significantly extended by analysis of cancer patient-derived CD8+ T cells revealing that by proteasome-catalyzed peptide splicing (PCPS) proteasomes can also fuse excised peptide fragments in a reverse proteolysis reaction, thereby generating new immune reactive spliced epitopes (splicetopes) with an amino acid sequence that differs from that of the substrate protein (*Hanada et al., 2004*; *Vigneron et al., 2004*). The isolation of splicetope-specific CD8+ T cells from cancer patients and the finding that splicetope-specific CD8+ T cells derived from tumor-infiltrating lymphocytes (TILs) inhibited the engraftment of human acute myeloid leukemia cells in SCID mice indicated the potential immune relevance of such tumor antigen-derived splicetopes (*Robbins et al., 1994*).

Importantly, for the fibroblast growth factor (FGF)-5 and several splicetopes derived from the tumor differentiation antigen gp100mel, in vitro proteasome splicing reactions were also found to mimic the in vivo splicing reactions (*Ebstein et al., 2016*; *Warren et al., 2006*), suggesting that in vitro PCPS reactions may be a useful tool to discover new spliced epitopes generated from tumor antigens of interest. To be able to identify splicetopes in in vitro PCPS experiments independent of the availability of patient-derived CD8+ T cells, we developed the prediction algorithms ProteaJ (*Liepe et al., 2010*) and the here-described ProtAG. Using these algorithms, we established an inclusion list of potentially immune-relevant spliced peptides theoretically generated from a given antigen, which in combination with the mass spectrometric analysis of the in vitro digest should allow the identification of new splicetopes. Testing the feasibility of such an algorithm-aided 'reverse immunology' approach, we had isolated CD8+ T cells from *Listeria monocytogenes*–infected mice that specifically recognized two phospholipase PlcB-derived splicetopes generated by the proteasome in vitro and in vivo (*Platteel et al., 2017*).

Targeting somatic cancer-specific driver mutations derived from neoantigens by TCR-mediated adoptive T cell transfer (ATT) represents a promising approach for personalized cancer therapy (*Blankenstein et al., 2015*). One drawback of this approach is that often neoepitopes may not exhibit HLA class I binding affinities sufficient to trigger an efficient T cell response or not be generated efficiently by the proteasome. In fact, even if a suitable neoepitope is generated, its HLA haplotype specificity frequently does not match with the patient's HLA class I allele, consequently excluding these tumor patients from ATT.

As outlined above, taking advantage of PCPS for the identification of spliced neoepitopes (neo-splicetopes) may therefore represent an interesting approach to identify suitable TCR targets when the recurrent somatic mutations in a tumor antigen do not result in the production of a non-spliced tumor neoepitope either exhibiting a sufficient HLA class I binding affinity or the appropriate HLA class I haplotype. Furthermore, due to the ligation of two distant generated peptide fragments PCPS not only possesses the interesting potential to generate high-affinity neo-splicetopes harboring the respective somatic mutation but also to extend the HLA haplotype diversity of epitopes generated from a given neoantigen.

In a proof-of-principle 'reverse immunology' study, we here identified HLA-A*02:01 restricted putative neo-splicetopes predicted by spliced peptide prediction algorithm derived from the two recurrent somatic mutations KRAS$^{G12V}$ and RAC2$^{P29L}$. TCRs specific for the putative neo-splicetopes were generated in huTCR-α/huTCR-β gene loci transgenic HLA-A*02:01 mice (*Li et al., 2010*). TCRs recognized the respective putative neo-splicetope with high efficacy when tested in vitro. However, we failed to detect a neo-splicetope-specific T cell response when testing the in vivo (in cellulo) generation of the predicted neo-splicetopes and thus failed to gain evidence that the two KRAS$^{G12V}$ and RAC2$^{P29L}$-derived neo-splicetopes were also generated in vivo as predicted by algorithm-aided studies. In addition, only the predicted neo-splicetope for RAC2$^{P29L}$ could be confirmed by in vitro proteasomal digest. The experiments pose severe questions on the applicability of the previously highlighted pipeline (*Mishto et al., 2019*) for the identification of immune-relevant neo-splicetopes.

## Results

### Prediction of KRAS$^{G12V}$-derived neo-splicetope

KRAS is one of the most frequently mutated genes in human cancer with the G12X (X = V, S, D, A, C) substitution accounting for most of the mutations found in this position. However, none of these mutations result in the formation of a non-spliced (linear) high-affinity HLA-A*02:01 binding

neoepitope (IC50 <100 nM). Since KRAS harbors the oncogenic mutation G12V in approximately 30% of pancreatic ductal adeno carcinoma and 20% of the colon and non-small cell lung cancers, we used the algorithm ProtAG for theoretical prediction of spliced peptides delineated from the $KRAS^{G12V}_{2-35}$ protein sequence. The algorithm-predicted spliced 9mer peptides were submitted to netMHCpan 4.0 (*Jurtz et al., 2017*) to screen for putative neo-splicetopes with predicted HLA-A*02:01 binding affinity IC50 <100 nM and carrying the mutant $V_{12}$ amino acid residue. Using this approach led to the identification of putative nonamer neo-splicetopes ($KRAS^{G12V}_{5-8/10-14}$, KLVV/GA$\underline{V}$GV, $IC_{50}$ 33.4; sp1), ($KRAS^{G12V}_{5-9/11-14}$, KLVV/A$\underline{V}$GV, $IC_{50}$ 37.50; sp2), and ($KRAS^{G12V}_{5-10/12-14}$, KLVVVG/$\underline{V}$GV, $IC_{50}$ 72.90; sp4), of which sp1 and sp2 allowed for TCR generation (*Blankenstein et al., 2019*).

## Generation of $KRAS^{G12V}$splicetope-specific TCRs in a humanized mouse model

To analyze the immunogenicity of spliced epitopes, we utilized transgenic mice (ABabDII mice) that harbor the human TCRαβ gene loci as a source for a diverse human TCR repertoire that is selected by chimeric HLA-A*02:01 (*Li et al., 2010*). Upon immunization with the peptides KLVVGA$\underline{V}$GV and KLVVVA$\underline{V}$GV (representing sp1 and sp2, respectively), these mice mounted a CD8$^+$ T cell response detected by in vitro re-stimulation of peripheral blood lymphocytes 7 days after the last immunization, whereas mice without immunization did not show reactivity (*Figure 1A*, *Figure 1—figure supplement 1A*). Both peptides induced a specific CD8$^+$ T cell response. By sorting IFNγ-positive sp1- and sp2-reactive CD8$^+$ T cells from splenocytes of responder mice using IFNγ-capture assay (not shown), specific TCRs were isolated upon rapid amplification of cDNA end (5′RACE)-PCR and cloning of the most abundant rearranged TCR-α and TCR-β genes for each individual mouse. One TCR directed against sp1 epitope (1376) and two TCRs specific for sp2 epitope (9383B2 and 9383B14) were isolated. Codon-optimized sequences encoding the α- and β-chains were linked with a P2A element and inserted into retroviral expression vector pMP71, transduced into human peripheral blood mononuclear cells (PBMC) (*Figure 1B*, *Figure 1—figure supplement 1B*) and tested for specificity measuring release of IFNγ in a co-culture with TAP-deficient T2 cells loaded with titrated amounts of sp1 (*Figure 1C*) or sp2 peptides (*Figure 1—figure supplement 1C*), respectively. All three TCRs induced robust IFNγ release at peptide concentrations of up to $10^{-10}$ M, suggesting high functional avidity for these TCRs. For $TCR_{1376}$ and $TCR_{9383B2}$, cross-reactivity to the in silico predicted linear $KRAS^{G12V}$ KLVVVGA$\underline{V}$GV peptide was only seen for the highest peptide concentrations (*Figure 1D* and not shown, respectively).

## $KRAS^{G12V}$splicetope-specific TCRs do not recognize cancer cells endogenously expressing mutant $KRAS^{G12V}$

One of the critical tests for the usefulness of therapeutic TCRs in genetically modified T cells is the recognition of cancer cells that endogenously express the respective mutation. This approach was even more decisive for our approach because so far the predicted neo-splicetopes had been predicted in silico but not detected in cells. Therefore, PBMC genetically engineered to express sp1- and sp2-specific TCRs were co-cultured with a series of cancer cell lines that harbored the G12V mutation within the KRAS gene. MCF7 and Mel624 cells with two KRAS wildtype copies served as controls. Whereas some of the cell lines used expressed HLA-A*02:01 (*Figure 2A*, *Figure 2—figure supplement 1A, B*), the HLA-A*02:01-negative cell lines were transduced with an HLA-A*02:01 expressing retroviral construct (*Figure 2B*). Presence of sufficient amounts of HLA-A*02:01 for T cell recognition was analyzed by prior loading of the tumor cells with $10^{-6}$ M of the respective peptide. In all cases, peptide-loaded cancer cells were recognized by $TCR_{1376}$ (*Figure 2A, B*, red bars) or $TCR_{9383B2}$ (*Figure 2—figure supplement 1A, B*, red bars) and $TCR_{9383B14}$ (*Figure 2—figure supplement 1A, B*, orange bars) transduced T cells, respectively. In contrast, IFNγ release by $TCR_{1376}$ (*Figure 2A, B*) or $TCR_{9383B2}$ and $TCR_{9383B14}$ engineered T cells (*Figure 2—figure supplement 1A, B*) was not above background when cancer cells were co-cultured without prior peptide loading, indicating that the endogenous $KRAS^{G12V}$ protein is not recognized. Cancer cells were also treated with IFNγ 48 hr prior to co-culture with the respective TCR-modified PBMCs. As exemplarily shown for sp2-specific TCRs, again only peptide-loaded tumor cells were recognized (*Figure 2—figure supplement 1B*).

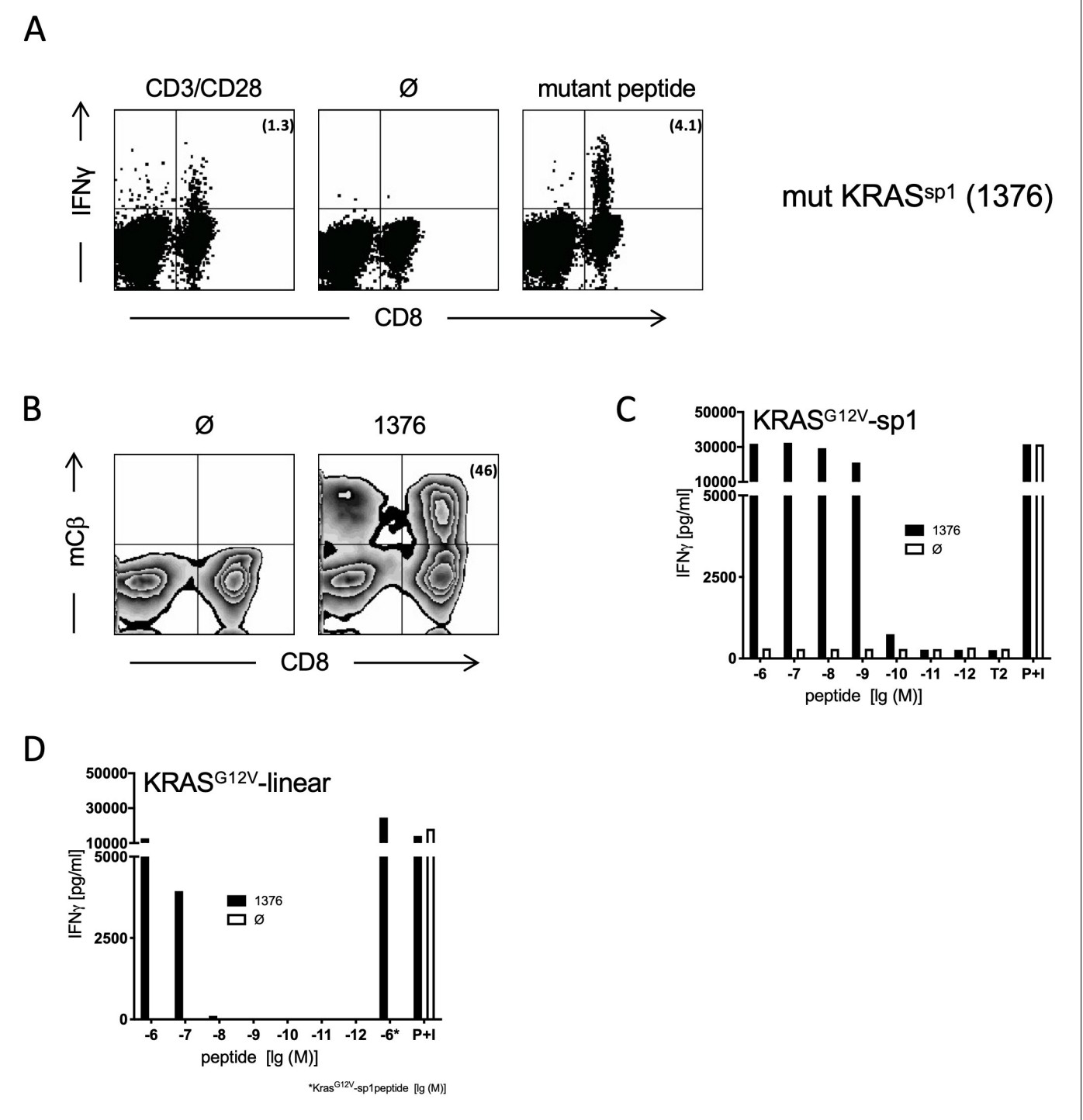

**Figure 1.** Generation and characterization of TCRs specific for spliced epitope 1 (sp1) of mutant KRAS[G12V]. (**A**) A representative example of ex vivo intracellular cytokine staining (ICS) analysis of KRAS mutant peptide immunized ABabDII mice (*Li et al., 2010*) 7 days after the last immunization with sp1 (KLVVGAVGV). Stimulation with CD3/CD28 beads served as positive control, co-culture without peptide (Ø) was used as negative control. Numbers in brackets represent percent IFNγ[+] CD8[+] T cells, respectively. Spleens of mice with IFNγ-reactive CD8[+] T cells were cultured for 10 days in the presence of $10^{-8}$ M of sp1 KRAS peptide, and reactive CD8[+] T cells were purified by IFNγ-capture assay for isolation of TCR α and β chains by RACE-PCR. (**B**) The corresponding TCR α and β chains isolated from one KRAS[G12V] sp1 peptide immunized ABabDII mouse, respectively (1376), were cloned into retroviral vector pMP71 and reexpressed in human PBMC. Transduction efficacy was measured by staining of the mouse TCRβ constant chain on CD8[+] T cells, and the number of positive CD8[+] T cells is shown in brackets. (**C**) TCR gene transfer confers specificity for mutant spliced KRAS[G12V] peptide KLVVGAVGV (sp1). IFNγ production of KRAS[G12V] splice-specific 1376 TCR-transduced T cells upon co-culture with sp1-peptide-loaded T2 cells
*Figure 1 continued on next page*

Figure 1 continued

(1376 [solid bars]). As negative control, T2 cells were not peptide loaded. For maximal stimulation, phorbol myristate acetate (PMA) and ionomycin (p + I) were added to the co-culture. All target cells were also co-cultured with non-transduced T cells (Ø, open bars). (D) TCR gene transfer confers cross-reactivity for mutant linear KRAS$^{G12V}$ peptide KLVVVGA_VGV. IFNγ production of KRAS$^{G12V}$ splice-specific 1376 TCR-transduced T cells upon co-culture with KRAS$^{G12V}$ linear peptide-loaded T2 cells (1376 [solid bars]). As negative control, T2 cells were not peptide loaded. For maximal stimulation, PMA and ionomycin (p + I) were added to the co-culture. All target cells were also co-cultured with non-transduced T cells (Ø, open bars). Experiments were done at least in duplicate.

The online version of this article includes the following figure supplement(s) for figure 1:

**Figure supplement 1.** Generation and characterization of TCRs specific for spliced epitope 2 (sp2) of mutant KRAS$^{G12V}$.

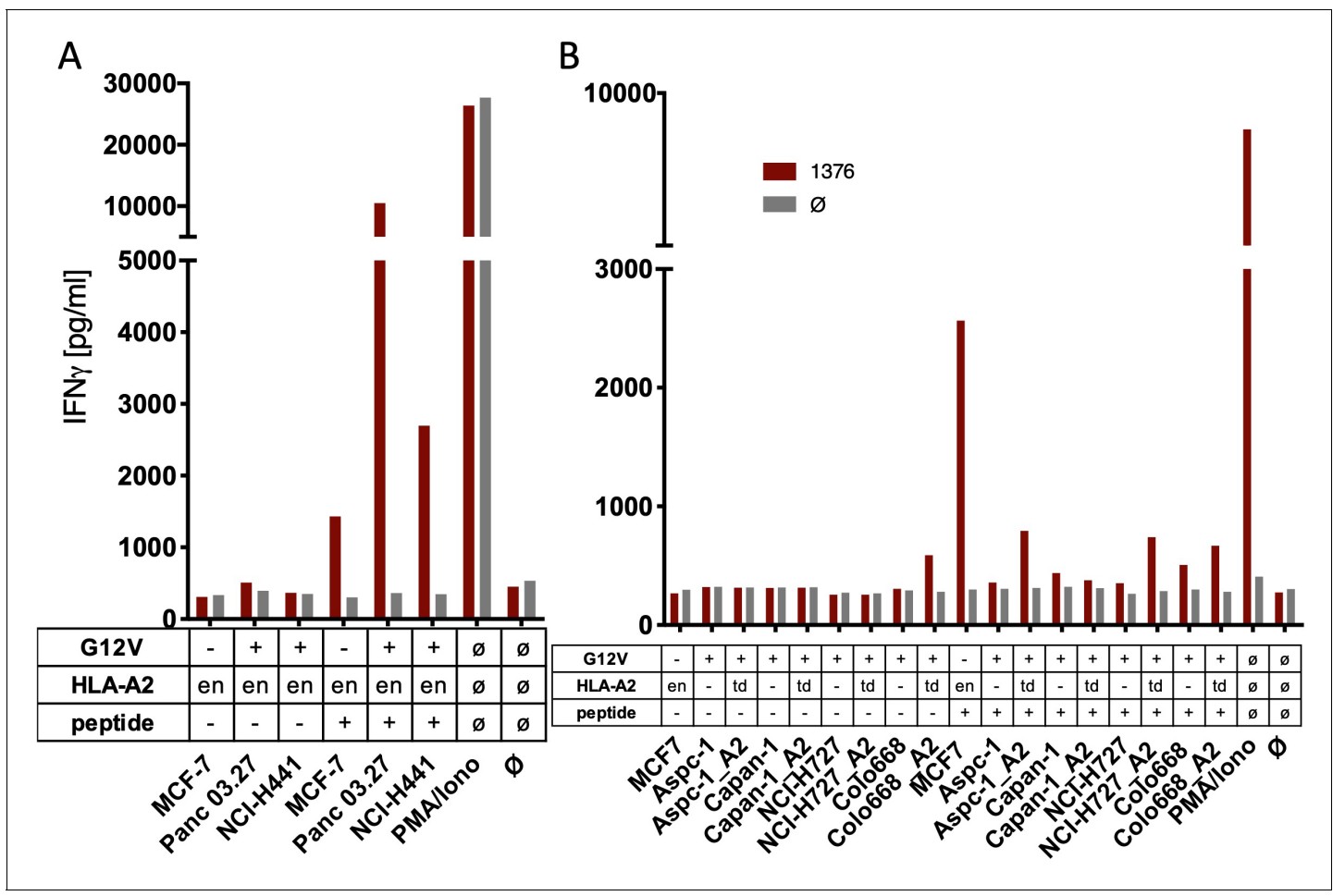

**Figure 2.** The spliced KRAS$^{G12V}$ epitope one is not recognized by spliced epitope 1 (sp1)-TCR-redirected T cells. (A) For analysis of natural processing and recognition of KRAS$^{G12V}$ epitopes, cell lines naturally expressing HLA-A2:01 and harboring the KRAS$^{G12V}$ mutation and HLA-A2:01$^{+}$ KRAS$^{wt}$ cell line MCF7 were co-cultured with KRAS$^{G12V}$ TCR$_{1376}$-redirected T cells. (B) HLA-A2:01-negative cell lines were transiently transduced with an HLA-A02:01 expressing retroviral construct (td) and co-cultured as in (A). IFNγ production of transduced T cells is shown (red bars). As positive control, peptide-loaded cells (+) were used, respectively. For maximal stimulation, phorbol myristate acetate (PMA) and ionomycin (PMA/Iono) were added, and all target cells were also co-cultured with non-transduced T cells (gray bars; Ø); en: endogenous expression of HLA-A2:01. Representative measurements are shown, and experiments were done at least in duplicate.

The online version of this article includes the following figure supplement(s) for figure 2:

**Figure supplement 1.** The spliced KRAS$^{G12V}$ epitope 1 (sp1) is not recognized by sp2-TCR-redirected T cells.

## T cells harboring KRAS$^{G12V}$splicetope-specific TCRs do not recognize overexpressed KRAS$^{G12V}$

One challenge of targeting neoepitopes with T cells is the low abundance of many neoantigens on the surface of the respective HLA class I molecules that may hamper recognition by T cells. To exclude low expression level as one reason for the failure of TCR$_{1376}$-engineered T cells to recognize the spliced form of the KRAS$^{G12V}$ peptide on the cancer cells, we generated cancer cells (MCF7, Mel624, and mouse NIH-HHD) that ectopically overexpressed triple minigenes encoding three copies of the KRAS$^{G12V}$ mutation interconnected by an AAY sequence that ensures proteasomal cleavage (*Spiotto et al., 2002*). We therefore generated triple minigene cassettes that either encoded the N-terminal 35mer polypeptide of KRAS$^{G12V}_{1-35}$ or as control triple minigenes that encoded the predicted non-spliced 10mer KRAS$^{G12V}_{5-14}$ peptide epitope, the spliced 9mer KRAS$^{G12V}_{5-8/10-14}$, or KRAS$^{wt}_{5-8/10-14}$ peptide epitope, respectively (*Figure 3A*). As shown in *Figure 3B–D*, TCR$_{1376}$-positive T cells efficiently recognized the KRAS$^{G12V}_{5-8/10-14}$ peptide when loaded either onto MCF7 (*Figure 3B*), Mel624 (*Figure 3C*), and mouse NIH-HHD (*Figure 3D*) cells or when expressed as a triple epitope. In contrast, no IFNγ release was elicited with cells expressing the triple KRAS$^{G12V}_{1-35}$ 35mer polypeptide (*Figure 3B–D*). Quantitative PCR analysis of KRAS$^{G12V}$ triple minigene 35mer and KRAS$^{G12V}$ triple epitope spliced nonamer revealed that KRAS$^{G12V}$ triple minigene 35mer is expressed almost twice as high as the KRAS$^{G12V}$ triple epitope spliced nonamer (*Figure 3E*). Altogether, this indicates that the spliced peptide, theoretically predicted, is either not generated in vivo or, despite the overexpression of the KRAS$^{G12V}_{1-35}$ substrate, is produced at amounts insufficient to be recognized by KRAS$^{G12V}_{5-8/10-14}$-specific high-affinity T cells.

## KRAS$^{G12V}$ splice peptide-specific TCR$_{1376}$ cross-reacts with HLA-C07 allele

We initially identified two cell lines with the G12V mutation (SW480 and SW620) that induced IFNγ release by TCR$_{1376}$-transduced PBMC upon co-culture (*Figure 3—figure supplement 1A*). Upon co-culture with a panel of lymphoblastoid B cell lines (BLCLs) that harbor a series of different HLA class I molecules, a test for potential HLA allo-reactivity that we routinely perform with TCRs obtained from ABabDII mice, we uncovered reactivity to several BLCLs (*Figure 3—figure supplement 1B*), all of them having in common the expression of HLA-C*07 (*Supplementary file 1*). Reanalysis of the tumor cell lines SW480 and SW620, which originated from the same patient, confirmed expression of HLA-C*07 molecule (*Supplementary file 1*). To finally prove HLA allo-reactivity of the TCR$_{1376}$ to HLA-C*07, we performed co-culture with the HLA-deficient myelogenous leukemia cell line K562 that had been transduced with HLA-C*07:01, HLA-C*07:02, and HLA-A*02:01 molecules, respectively. The experiments showed that K562-C*07:01 and K562-C*07:02 cell lines were recognized by three independent TCR$_{1376}$-transduced PBMC donors irrespective of loading with peptide sp1, whereas K562-A02:01 cells only induced IFNγ release when these cells were loaded with sp1 peptide prior to co-culture (*Figure 3—figure supplement 1C*). These results clearly indicate that the TCR$_{1376}$ directly recognizes members of the HLA-C*07 sub-family and/or peptides bound therein as well as sp1 peptide bound to HLA-A*02:01.

## Triple 35mer polypeptide of KRAS$^{G12V}_{1-35}$ minigenes are not immunogenic in vivo

In order to analyze whether triple 35mer polypeptide of KRAS$^{G12V}_{1-35}$ minigenes would induce a CD8$^+$ T cell response in vivo, we performed immunizations of ABabDII mice with an adenovirus expressing the N-terminal 35mer polypeptide of KRAS$^{G12V}_{1-35}$ as a triple minigene. Despite multiple immunizations, neither restimulation with the linear 10mer KLVVGA<u>V</u>GV nor with the spliced epitopes sp1 KLVVGA<u>V</u>GV, sp2 KLVVVAVGV, sp3 YLVVVGAVGV, or sp4 KLVVVG<u>V</u>GV induced IFNγ release by CD8$^+$ T cells in an intracellular cytokine staining of PBMCs 7 days after the last immunization (*Figure 3—figure supplement 2*). This supports the notion that the KRAS$^{G12V}$ mutation is not immunogenic in the context of HLA-A*02:01, irrespective of whether splicing events occur or not.

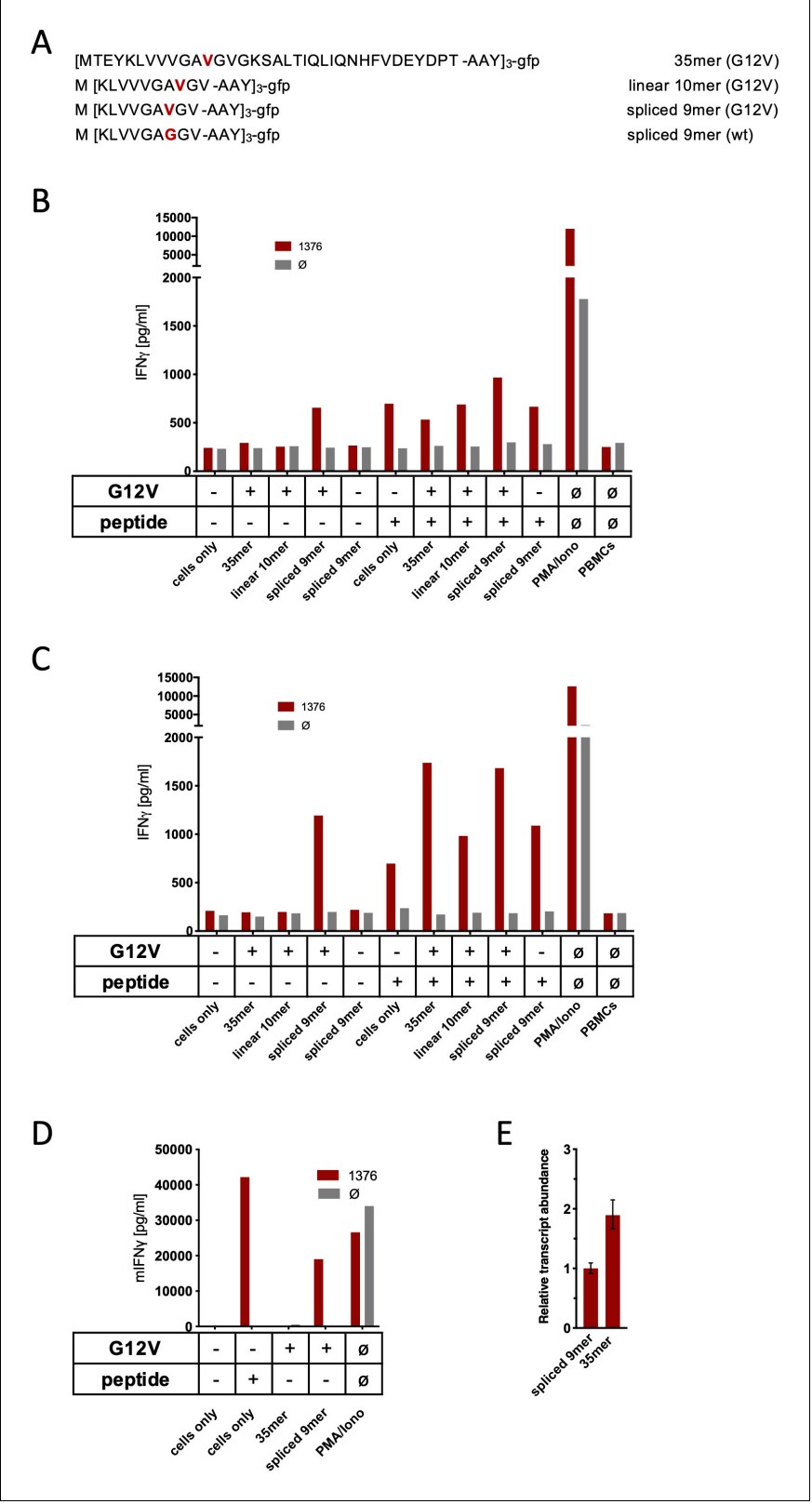

**Figure 3.** Co-culture of KRAS[G12V] splice-specific TCR (TCR[1376]) with human and mouse cells expressing KRAS[G12V] cDNA or triple epitopes. (**A**) Schematic representation of KRAS[G12V/wt] triple epitopes used for recombinant overexpression in MCF7, 624Mel, and NIH-HHD cells. TCR[1376] was retrovirally transduced into human PBMCs or TCR1xCD45.1xRag1[-/-] mouse splenocytes, respectively, and 10[4] transduced cells were co-cultured 1:1 with target

*Figure 3 continued on next page*

*Figure 3 continued*

cells (B: MCF7; C: 624Mel; D: NIH-HHD). Target cells were loaded with $10^{-6}$ M spliced peptide or transduced with either $KRAS^{G12V}$ triple minigene 35mer or $KRAS^{G12V}$ triple epitope spliced nonamer. $KRAS^{wt}$ triple epitope spliced nonamer and $KRAS^{G12V}$ triple epitope linear decamer were used as control. IFNγ production of transduced T cells is shown (red bars). For maximal stimulation, phorbol myristate acetate (PMA) and ionomycin (PMA/Iono) were added, and all target cells were also co-cultured with non-transduced T cells (gray bars; Ø). Representative measurements are shown, and experiments were done at least in duplicate. (E) Relative amounts of $KRAS^{G12V}$ triple minigene 35mer and $KRAS^{G12V}$ triple epitope spliced nonamer were determined by qPCR on transduced NIH-HHD cells. $KRAS^{G12V}$ triple epitope spliced nonamer expression is arbitrarily set to 1.

The online version of this article includes the following figure supplement(s) for figure 3:

**Figure supplement 1.** Co-culture of $KRAS^{G12V}$ splice peptide-specific TCR ($TCR_{1376}$) cross-reacts with HLA-C07 allele.

**Figure supplement 2.** Triple $KRAS^{G12V}_{1-35}$ minigene immunization does not generate cytotoxic T lymphocyte (CTL) response against predicted linear or spliced HLA-A02:01 epitopes.

**Figure supplement 3.** Base peak chromatogram of the synthetic polypeptides (A) $KRAS^{G12V}_{2-35}$ and (B) $KRAS^{G12V}_{2-32}$.

**Figure supplement 4.** Generation of the predicted putative $KRAS^{G12V}$-derived nonamer neo-splicetopes sp1, sp2, and sp4 from the synthetic polypeptide substrate $KRAS^{G12V}_{2-14}$ in a kinetic proteasome-catalyzed peptide splicing experiment.

**Figure supplement 5.** Generation of the predicted putative $KRAS^{G12V}$-derived nonamer neo-splicetopes sp1 and sp4 from the synthetic polypeptide substrate $KRAS^{G12V}_{2-21}$ in a kinetic proteasome-catalyzed peptide splicing experiment.

## Predicted $KRAS^{G12V}$ spliced peptides are not generated in in vitro PCPS reactions

Our failure to detect immune-reactive $KRAS^{G12V}$-derived neo-splicetopes under in vivo conditions raised doubts with respect to the reliability of the previously proposed solely algorithm-based pipeline for identification of immune-relevant neo-splicetopes (*Mishto et al., 2019*). Therefore, we studied the generation of the $KRAS^{G12V}$-derived neo-splicetopes in more detail in in vitro PCPS assays. Accordingly, the polypeptide substrates $KRAS^{G12V}_{2-35}$, $KRAS^{G12V}_{2-32}$, $KRAS^{G12V}_{2-21}$, and $KRAS^{G12V}_{2-14}$ were synthesized. However, due to the extreme hydrophobicity of the KRAS protein, the designed longer polypeptide substrates $KRAS^{G12V}_{2-35}$ and $KRAS^{G12V}_{2-32}$ encountered considerable difficulties during synthesis and subsequent purification, resulting in a highly impure product not suited for in vitro digestion experiments (*Figure 3—figure supplement 3*). Consequently, we used the polypeptides $KRAS^{G12V}_{2-21}$ and $KRAS^{G12V}_{2-14}$ for the in vitro PCPS reactions. $KRAS^{G12V}_{2-14}$ was chosen based on previous data showing that C-terminal cleavage generating the C-terminal anchor residue is not essentially required to generate a spliced gp100-derived epitope (*Liepe et al., 2010*; *Vigneron et al., 2004*). Monitoring the kinetics of proteasomal spliced peptide generation represents an essential parameter for assessing the fidelity of in vitro PCPS reactions. To search for spliced peptides, a fasta data file generated with ProtAG was loaded onto PD2.1 and the kinetics analyzed with LC Quan 2.7 (*Willimsky et al., 2021*). At t = 0, none of the predicted spliced neo-splicetopes was identified. However, following the generation of the $KRAS^{G12V}$-derived putative spliced neoepitopes from the polypeptide substrates $KRAS^{G12V}_{2-14}$ and $KRAS^{G12V}_{2-21}$ over time in in vitro PCPS reactions (*Figure 3—figure supplements 4* and *5*), all three predicted 9mer $KRAS^{G12V}$ spliced peptides ($KRAS^{G12V}_{5-8/10-14}$ sp1, $KRAS^{G12V}_{5-9/11-14}$, sp2, $KRAS^{G12V}_{5-10/12-14}$, sp4) were found to be generated from the $KRAS^{G12V}_{2-14}$ substrate (*Figure 3—figure supplement 4*). This corroborated previous findings of *Mishto et al., 2019*, who reported the in vitro generation of $KRAS^{G12V}_{5-8/10-14}$ and $KRAS^{G12V}_{5-9/11-14}$ from a longer $KRAS^{G12V}_{2-35}$ polypeptide substrate. Using the longer $KRAS^{G12V}_{2-21}$ polypeptide substrate for the in vitro PCPS reactions, only the putative neo-splicetopes $KRAS^{G12V}_{5-8/10-14}$ and $KRAS^{G12V}_{5-10/12-14}$ were generated (*Figure 3—figure supplement 5A*). The apparent contradiction between our in vivo experiments reported above and the results of the in vitro PCPS reactions was unexpected, considering that for the several spliced epitopes published so far there seemed to be a good correlation between the in vitro and in vivo results (*Dalet et al., 2011*; *Ebstein et al., 2016*; *Michaux et al., 2014*; *Mishto et al., 2012*; *Platteel et al., 2017*).

This led us to perform a more detailed MS analysis of the polypeptide substrate used for the in vitro PCPS experiments. Indeed, we found that most likely the accumulation of hydrophobic amino

acid residues within the KRAS$^{G12V}$ polypeptide substrates had led to mistakes during polypeptide synthesis, resulting in the synthesis of faulty polypeptides (*Supplementary file 2*) mimicking in sequence the results of the predicted splicing reaction (*Figure 3—figure supplement 5B*, *Willimsky et al., 2021*). Therefore, it was impossible to decide whether the candidate KRAS$^{G12V}$-derived spliced peptides identified in vitro were true splicing products or as it appeared the product of normal proteasomal cleavage of already preexisting faulty polypeptides inappropriately simulating a splicing event. Furthermore, depending on the substrate, in vitro generation of non-spliced epitopes can be by orders more efficient than the generation of spliced epitopes (*Mishto et al., 2019*). Thus, polypeptide substrates with mistakes in their sequence that are degraded at a rate similar to the rate of the correct substrate (*Figure 3—figure supplement 5C*) may become a prevalent source for the generation of faked spliced peptides. We eventually obtained a KRAS$^{G12V}_{1-21}$ polypeptide substrate (JPT Peptide Technologies, Berlin, Germany) without contaminants mimicking the predicted KRAS$^{G12V}_{5-8/10-14}$ (sp1) and KRAS$^{G12V}_{5-10/12-14}$ (sp4) splicing events. However, using this new KRAS$^{G12V}_{1-21}$ polypeptide as substrate for kinetic in vitro PCPS experiments, we now failed to identify generation of either predicted KRAS$^{G12V}$ 9mer neo-splicetope. Although our experiments cannot completely exclude the generation of minor amounts of KRAS$^{G12V}$-derived spliced peptides, they are in line with our failure to detect any immune-reactive KRAS$^{G12V}$-derived spliced epitopes in vivo.

## Identification and functional characterization of RAC2$^{P29L}$-derived neo-splicetopes

RAC2 is a small GTPase belonging to the Rho family of GTPases. The RAC2$^{P29L}$ mutation is another so-called driver mutation facilitating tumor growth as well as metastasis and thus presents a potential target in ATT. The linear RAC2$^{P29L}$ FLGEYIPTV epitope has been predicted with an IC$_{50}$ of 2 nM. To identify RAC2$^{P29L}$-specific neo-splicetopes, we applied the ProtAG algorithm to predict all theoretically possible RAC2$^{P29L}_{20-44}$-derived spliced peptides. From this initial screen, we selected all theoretical linear spliced 9mer peptides with a calculated HLA-A*02:01 binding affinity of IC$_{50}$ < 100 nM (*Jurtz et al., 2017*). To establish a cleavage map and identify all linear proteasomal cleavage products generated from the RAC2$^{P29L}_{20-44}$ polypeptide substrate, we performed in vitro digestions for 24 hr and 48 hr using erythrocyte and LcL 20S proteasomes (*Willimsky et al., 2021*). In these digests, also the non-spliced RAC2$^{P29L}_{28-36}$ neoepitope FLGEYIPTV was identified (*Figure 4A*). To search for spliced peptides, a fasta data file generated with ProtAG was loaded onto PD2.1 (*Willimsky et al., 2021*). In this search, the spliced RAC2$^{P29L}_{28-34/36-37}$ ($_{28}$FLGEYIP$_{34}$/$_{36}$VF$_{37}$) peptide, with a calculated HLA-A*02:01 binding affinity of IC$_{50}$ = 24,7 nM, was found to be the only HLA-A*02:01 restricted putative RAC2$^{P29L}$ neo-splicetope with an IC$_{50}$ < 100 nM generated. To confirm the initial identification of RAC2$^{P29L}_{28-34/36-37}$ kinetic in vitro, PCPS reactions were performed and analyzed by applying the LC Quan software version 2.5 (Thermo Fisher) (*Figure 4B*, *Figure 4—figure supplement 1*).

The amounts of peptides generated in an in vitro processing reaction can vary dramatically depending on the assay conditions allowing only a relative estimation. However, judging by ion counts in vitro generation of the non-spliced RAC2$^{P29L}_{28-36}$ neoepitope was approximately 200-fold more efficient than generation of the putative neo-splicetope RAC2$^{P29L}_{28-34/36-37}$ (*Figure 4A*).

To exclude that generation of RAC2$^{P29L}_{28-34/36-37}$ was the result of an accidental singular splicing event we screened the digests for additional PCPS products. Interestingly, the putative RAC2$^{P29L}_{28-34/36-37}$ neo-splicetope seemed to be the result of the excision of a single aa residue (T$_{35}$) and the C-terminal ligation of the dipeptide $_{36}$VF$_{37}$ to the N-terminal $_{28}$FLGEYIP$_{34}$ fragment. However, repetitive specific ligation of a dipeptide in a PCPS reaction would require the unlikely existence of a corresponding specific dipeptide binding site close to the active site and the respective acceptor fragment.

Therefore, we hypothesized that generation of the RAC2$^{P29L}_{28-34/36-37}$ spliced peptide was the result of a multistep process involving a larger already spliced precursor product. Indeed, as shown in *Figure 4B* in in vitro kinetic experiments and by detailed mass spectrometric analyses, we identified the RAC2$^{P29L}_{28-34/36-40}$ FLGEYIP/VFDNY polypeptide being the largest already spliced epitope precursor peptide. Supporting that RAC2$^{P29L}_{28-34/36-37}$ was generated via precursor peptides, we also detected the corresponding N-terminal splice acceptor peptide RAC2$^{P29L}_{28-34}$ (FLGEYIP) and the C-terminal splice donor peptide RAC2$^{P29L}_{36-40}$ (VFDNY) (*Figure 4C*). This suggests that

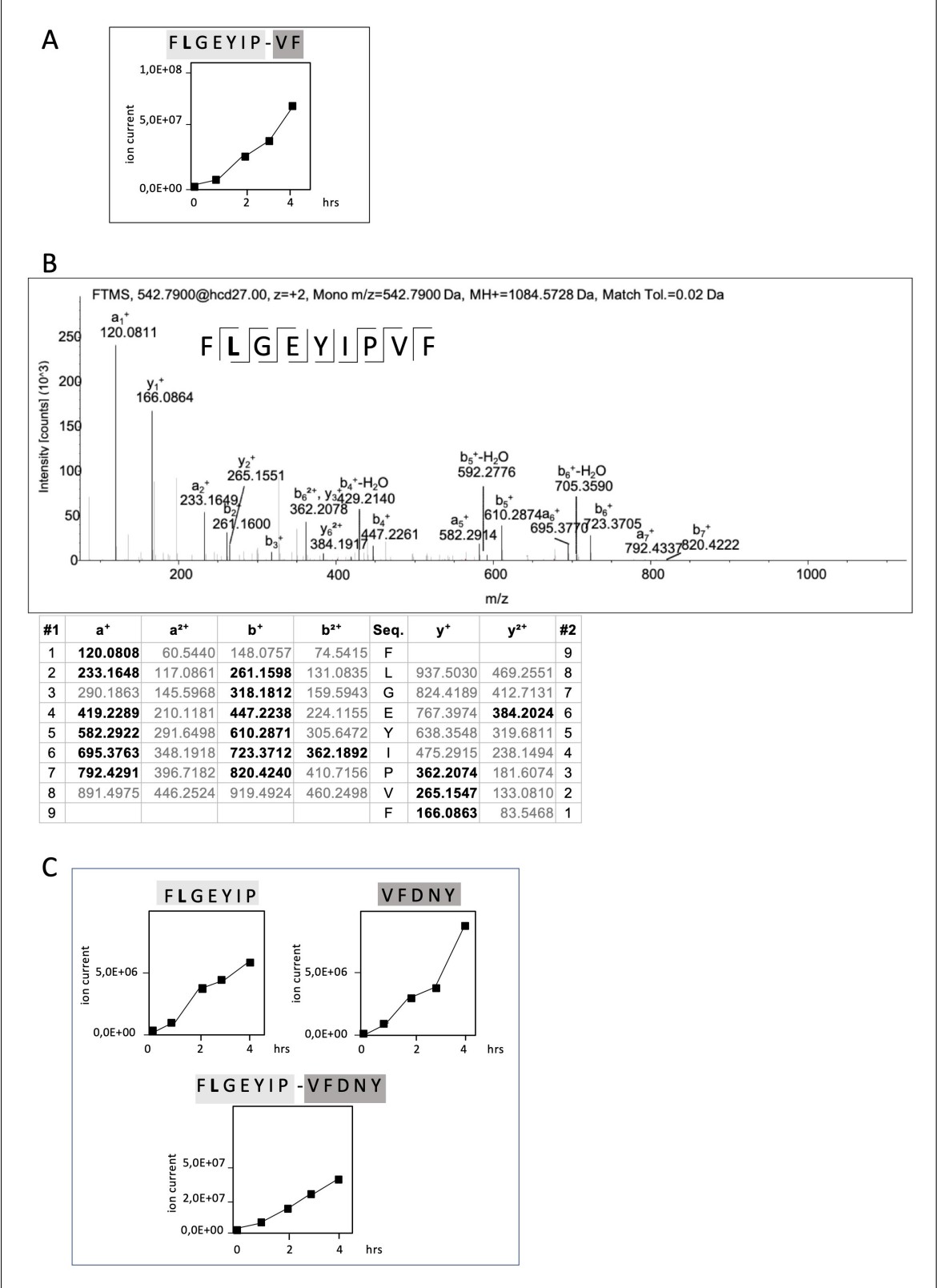

**Figure 4.** Non-spliced and spliced peptides generated from RAC2$^{P29L}_{20–44}$ in kinetic proteasome-catalyzed peptide splicing reactions. The candidate RAC2$^{P29L}$ neo-splicetope is generated via a C-terminally extended precursor peptide. (**A**) Kinetics of the generation of the 9mer candidate RAC2$^{P29L}_{28-34/36-37}$ neo-splicetope and the non-spliced RAC2$^{P29L}_{28-36}$ neoepitope. Note that generation of the non-spliced RAC2$^{P29L}_{28-36}$ peptide is significantly more efficient than the generation of the spliced RAC2$^{P29L}_{28-34/36-37}$. (**B**) MS/MS spectra of the candidate RAC2$^{P29L}_{28-34/36-37}$ neo-splicetope. (**C**) Kinetics

*Figure 4 continued on next page*

Figure 4 continued

of the generation of the non-spliced acceptor FLGEYIP and donor VFDNY peptides and the generation of the C-terminally extended spliced precursor peptide RAC2$^{P29L}$ $_{28-34/36-40}$ FLGEYIP-VFDNY. The MS/MS spectra for the identified RAC2$^{P29L}$-derived peptides are shown in **Figure 4—figure supplement 1**.

The online version of this article includes the following figure supplement(s) for figure 4:

**Figure supplement 1.** MS/MS spectra of the identified RAC2$^{P29L}$-derived peptides.

generation of the final RAC2$^{P29L}$$_{28-34/36-37}$ neo-splicetope requires an additional C-terminal proteasomal cleavage step for spliced epitope liberation (**Figure 4A**).

## RAC2$^{P29L}$$_{28-34/36-37}$ splicetope-specific TCR does not recognize RAC2$^{P29L}$ triple epitope 45mer

Spliced RAC2$^{P29L}$$_{28-34/36-37}$ peptide-specific TCRs were generated by immunizing ABabDII mice with the corresponding synthetic 9mer peptides (not shown). For analysis of in vivo generation and presentation of the spliced RAC2$^{P29L}$$_{28-34/36-37}$ peptide, we transduced Mel21a cells to express a triple RAC2$^{P29L}$$_{1-45}$ 45mer polypeptide minigene (**Figure 5A**) and monitored HLA-A*02:01 epitope presentation by T cell recognition. As shown in **Figure 5B**, no IFNγ release was obtained for the putative neo-splicetope RAC2$^{P29L}$$_{28-34/36-37}$ using TCR$_{20967A2}$-transduced T cells, while peptide-loaded Mel21a cells were readily recognized and IFNγ production demonstrated the target specificity of the TCR. Thus, despite the overexpression of the RAC2$^{P29L}$$_{1-45}$ 45mer substrate peptide, we failed to verify the in vivo generation of the RAC2$^{P29L}$$_{28-34/36-37}$ peptide. We also raised a TCR (TCR$_{22894}$) in ABabDII mice against the linear RAC2$^{P29L}$ peptide. Because Mel21a cells, transduced to express a triple RAC2$^{P29L}$$_{1-45}$ 45mer (**Figure 5B**) or RAC2$^{P29L}$ cDNA (**Figure 5C**), were readily recognized by T cells transduced with TCR$_{22894}$, we could exclude that our inability to detect cell surface expression of the RAC2$^{P29L}$$_{28-34/36-37}$ peptide was due to defects in the antigen presentation pathway. We repeated the experiments with TCR-engineered mouse T cells derived from TCR1xCD45.1xRag1$^{-/-}$ mouse splenocytes (expressing a monoclonal irrelevant TCR against SV40 large T) to monitor RAC2$^{P29L}$$_{28-34/36-37}$ peptide cell surface expression using mouse NIH-HHD cells expressing a chimeric HLA-A02:01 (HHD) molecule. As observed in Mel21a cells, the non-spliced RAC2$^{P29L}$ neoepitope (derived from RAC2$^{P29L}$$_{1-45}$ 45mer as well as cDNA) was efficiently presented also by NIH-HHD cells, excluding potential differences in the catalytic properties of mouse and human proteasomes (**Figure 5D**). More importantly, again the spliced peptide-specific TCR$_{20967A2}$ did not confer any reactivity to T cells upon co-culture without prior peptide loading of the target cells or overexpression of the spliced epitope (**Figure 5B–D**). Quantitative PCR analysis of the triple RAC2$^{P29L}$$_{1-45}$ 45mer polypeptide minigene and RAC2$^{P29L}$ triple epitope spliced nonamer expressed in mouse NIH-HHD cells revealed that RAC2$^{P29L}$$_{1-45}$ 45mer polypeptide minigene is expressed almost fivefold higher than the RAC2$^{P29L}$ triple epitope spliced nonamer (**Figure 5E**). These experiments do not categorically exclude any in vivo generation of the in vitro identified RAC2$^{P29L}$-derived spliced peptide. However, they clearly demonstrate that even if the RAC2$^{P29L}$$_{28-34/36-37}$ neo-splicetope is derived from an overexpressed substrate protein, its amounts are negligible and insufficient to allow its recognition by T cells.

In summary, our results strongly question the idea that in vitro PCPS reaction simulates the in vivo situation with the same high fidelity as the in vitro generation of non-spliced epitopes and contradicts the previously highlighted idea that an algorithm-supported identification of in vitro-generated spliced epitopes is a suitable general approach for the facilitated identification of tumor-specific immune-relevant neo-splicetopes for consecutive TCR generation.

## Discussion

Effective CD8$^+$ T cell-induced immune responses depend on both the quality and the amount of proteasome-generated antigenic peptides available for presentation by HLA class I molecules to peptide-specific TCR at the cell surface (**Niedermann et al., 1999**; **Princiotta et al., 2003**). Not neglecting TCR affinity or the HLA class I binding affinity of an epitope, in each case the amount of a

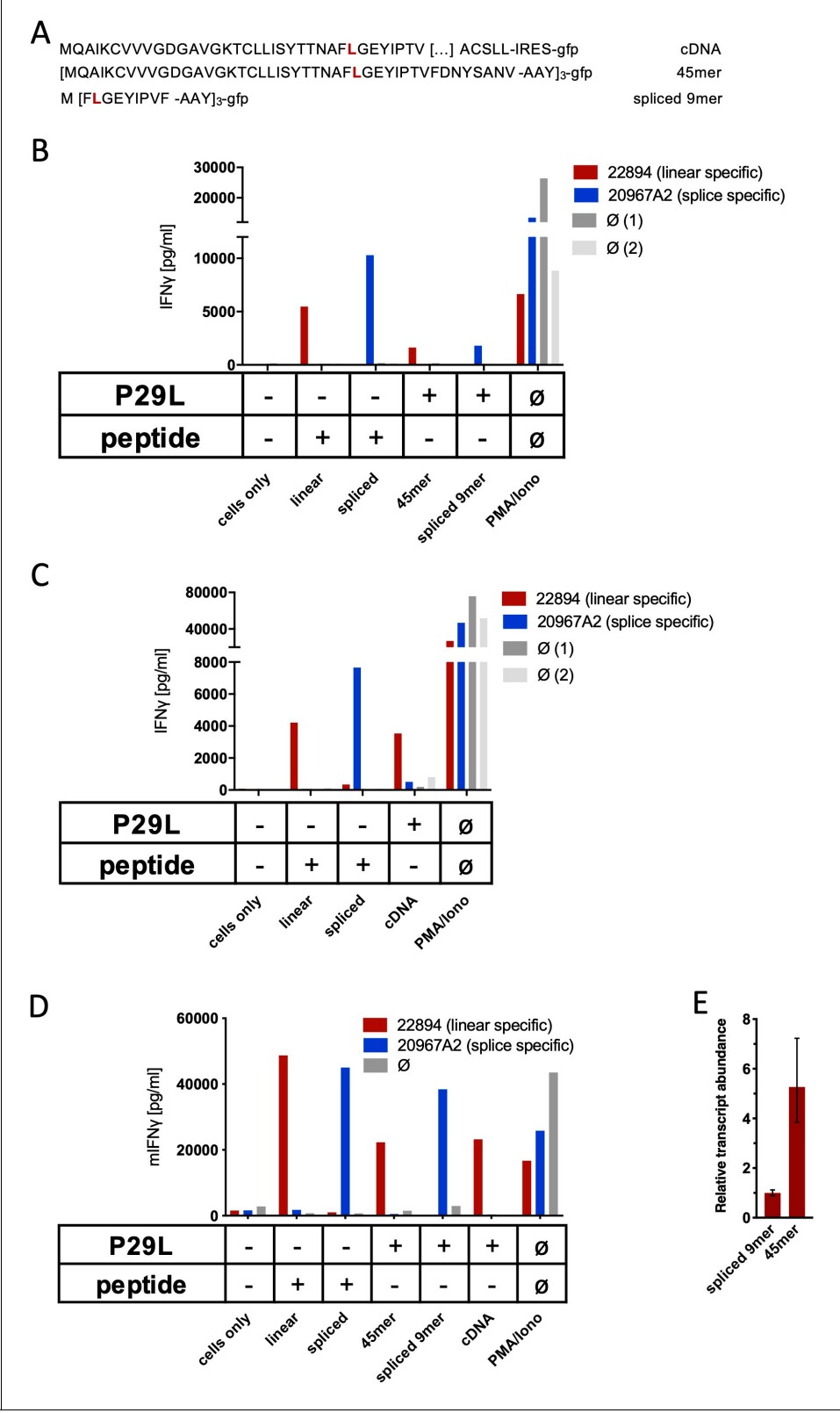

**Figure 5.** Co-culture of RAC2[P29L] linear-specific TCR (TCR[22894]) and Rac2[P29L] splice-specific TCR (TCR[20967A2]) with cells expressing Rac2[P29L] triple epitopes and cDNA. (**A**) Schematic representation of RAC2[P29L] cDNA and triple epitopes used for recombinant overexpression in Mel21a and NIH-HHD cells. (**B, C**) TCRs were retrovirally transduced into human PBMCs and $10^4$ transduced cells were co-cultured 1:1 with Mel21a target cells. (**D**) TCRs were retrovirally transduced into TCR1xCD45.1xRag1[-/-] mouse splenocytes, and $10^4$ transduced cells were co-cultured 1:1 with NIH-HHD target cells.

*Figure 5 continued on next page*

*Figure 5 continued*

Respective human and mouse target cells were loaded with $10^{-6}$ M spliced or non-spliced RAC$^{P29L}$ peptide, or transduced with either Rac2$^{P29L}$ triple epitope 45mer, Rac2$^{P29L}$ triple epitope nonamer, or Rac2$^{P29L}$ cDNA. Upon co-culture with recombinant TCR$^+$ T cells, IFN$\gamma$ release was measured. For maximal stimulation, phorbol myristate acetate (PMA) and ionomycin (PMA/Iono) were added, and all target cells were also co-cultured with non-transduced T cells (gray bars; $\emptyset$). Representative measurements are shown, and experiments were done at least in duplicate. (E) Relative amounts of Rac2$^{P29L}$ triple epitope 45mer and Rac2$^{P29L}$ triple epitope nonamer were determined by qPCR on transduced NIH-HHD cells. Rac2$^{P29L}$ triple epitope nonamer expression is arbitrarily set to 1.

specific epitope generated by proteasomes from a given antigen has to rise above a certain threshold to elicit a relevant T cell response.

In addition to substrate amounts and protein turnover rates, epitope generation efficiency of both non-spliced and spliced epitopes is determined by the sequence of the epitope, its surrounding protein sequence, and connected with the cleavage site usage and cleavage strength of proteasomes (*Mishto et al., 2014*; *Niedermann et al., 1995*; *Sijts and Kloetzel, 2011*). Thus, even high-affinity peptides, if embedded in a non-favorable protein sequence, will not surpass the necessary threshold for eliciting a T cell response.

The cleavage properties of proteasomes have been intensively studied. However, due to the complexity of protein sequences and lacking information on cleavage strength that decisively determines epitope generation efficiency, algorithms predicting proteasomal generation of immune-relevant non-spliced epitopes still do not reach prediction efficiencies sufficient for large-scale 'reverse immunology' approaches (*Calis et al., 2015*; *Di Carluccio et al., 2018*; *Singh and Mishra, 2016*). Thus, many predicted epitopes may be false-positives, which could impede immunotherapy, for example, neoantigen vaccines. Prediction algorithms for spliced epitopes, which are based on protein sequence and proteasomal cleavage properties, do not even yet exist.

Therefore, in vitro generation of epitopes using purified 20S proteasomes and synthetic polypeptide substrates encompassing the epitope(s) of interest, in combination with mass spectrometric analyses and both in vitro and in vivo CD8$^+$ T cell assays, still represents the most frequently used tool to validate the generation of immune-relevant non-spliced peptides, assuming that it closely simulates the in vivo (*in cellulo*) situation with respect to both quality and relative amounts of the epitope (*Kessler et al., 2001*; *Kessler and Melief, 2007*; *Sijts and Kloetzel, 2011*).

While a number of virus- or tumor-derived non-spliced epitopes have been validated by in vitro experiments and correlated to the in vivo situation, examples for spliced HLA class I epitopes are still very limited. Nevertheless, what applied to non-spliced epitopes also seemed to be valid for spliced HLA class I epitopes generated in vitro by PCPS. Thus, FGF-5, SP110, and several gp100-derived spliced epitopes that are recognized by CD8$^+$ T cells on the cell surface were demonstrated to be produced also in in vitro PCPS assays (*Ebstein et al., 2016*; *Vigneron et al., 2019*). Although in these cases the generation efficiency of spliced epitopes in in vitro PCPS assays seemed to be in line with the in vivo situation, it should be noted, however, that the abundance of spliced epitopes presented at the cell surface is a matter of substantial controversy (*Liepe et al., 2016*; *Liepe et al., 2019*; *Mylonas et al., 2018*; *Paes et al., 2019*; *Rolfs et al., 2019*). Thus, in light of recent reports (*Mylonas et al., 2018*; *Rolfs et al., 2019*) the amount of cell surface-presented spliced epitope seems to be considerably less than initially estimated.

On the other hand and supporting a potential immune relevance, the initial discovery of the splicing event and spliced epitopes was based on the identification of patient-derived CD8$^+$ T cells reactive towards spliced epitopes generated from tumor antigens (*Hanada et al., 2004*; *Vigneron et al., 2004*). Widespread identification of spliced epitopes is however limited by the rare availability of corresponding specific CD8$^+$ T cells. Therefore, we developed prediction algorithms allowing the mass spectrometric identification of predicted and in vitro proteasome-generated spliced peptides. Indeed, applying such an algorithm-aided 'reverse immunology' approach successfully led previously to the identification of two spliced phospholipase PlcB epitopes that primed antigen-specific CD8$^+$ T cells in *L. monocytogenes*-infected mice (*Platteel et al., 2017*).

Because somatic mutations in tumor antigens frequently do not result in the generation of neoepitopes suitable for generation of TCRs for ATT therapy, we applied spliced peptide prediction algorithms to identify neo-splicetopes with HLA-A*02:01 binding affinity predicted to be generated from the mutant tumor antigens KRAS$^{G12V}$ and RAC2$^{P29L}$ and used those for TCR generation.

In vitro PCPS experiments in combination with MS analysis aiming at the identification of the algorithm-predicted putative KRAS$^{G12V}$-derived neo-splicetopes, however, gave no final evidence for their in vitro generation. In the initial kinetic splicing reactions, we seemed to have identified the predicted spliced peptides, thereby corroborating also data obtained with a KRAS$^{G12V}_{2-35}$ polypeptide substrate reported by *Mishto et al., 2019*.

However, we found that most likely the extreme hydrophobicity of the KRAS amino acid composition had led to faulty polypeptide synthesis, resulting in polypeptide substrates mimicking in sequence the results of the predicted splicing reaction (*Figure 3—figure supplement 5*, *Supplementary file 2*). Considering that in general the generation of spliced peptides is significantly less efficient than that of non-spliced peptides (*Figure 4A*), even minor amounts of faulty peptide substrates will become prevalent in in vitro splicing reactions (see also *Figure 3—figure supplement 5B*). Because high-quality peptide synthesis reaches a purity of 95–99% at most, a thorough substrate analysis appears essential to avoid false-positive results in in vitro PCPS experiments, particularly for chemically difficult substrates. However, when we used a newly synthesized KRAS$^{G12V}$ substrate not contaminated with peptides mimicking the KRAS$^{G12V}_{5-8/9-14}$ and KRAS$^{G12V}_{5-10/12-14}$ splicing reactions, we failed to identify the in silico-predicted neo-splicetope. This negative result cannot finally prove the non-existence of the KRAS$^{G12V}_{5-8/9-14}$ and KRAS$^{G12V}_{5-10/12-14}$, but being in line with the in vivo experiments, one has to conclude that if these KRAS$^{G12V}$-derived spliced epitopes are generated, then their amount is below detectable level. In contrast to the negative results obtained with respect to KRAS$^{G12V}$, the analysis of RAC2$^{P29L}$ led to the identification of the in silico-predicted RAC2$^{P29L}_{28-34/36-37}$ peptide in in vitro PCPS experiments. Nevertheless, the putative RAC2$^{P29L}_{28-34/36-37}$ neo-splicetope was generated significantly less efficient than the non-spliced RAC2$^{P29L}_{28-36}$ neoepitope.

Testing the in vivo generation of the spliced KRAS$^{G12V}_{5-8/10-14}$ and RAC2$^{P29L}_{28-34/36-37}$ peptides using the respective peptide-specific TCRs, which were of high affinity and recognized as little as $10^{-10}$ M peptide, we obtained no T cell signal and no evidence for the immune relevance of the two candidate neo-splicetopes, independent of the experimental conditions. Our experiments provide no evidence that either KRAS$^{G12V}_{5-8/10-14}$ or RAC2$^{P29L}_{28-34/36-37}$ are produced in vivo or presented at the cell surface. However, even in case both neo-splicetopes were generated in vivo, their generation efficiency and the total amount presented by HLA-A*02:01 molecules on the cell surface are too low to be of any immune significance. One possible explanation for the failure to verify the in vitro PCPS reaction for RAC2$^{P29L}_{28-34/36-37}$ in in vivo settings could be the high substrate and proteasome concentration as used for in vitro PCPS, thereby forcing splicing reactions that do either not or only inefficiently occur under in vivo conditions where only a single substrate protein enters the catalytic cavity of the proteasome for processing at a given time.

Thus, quite in contrast to the experience resulting from proteasome-dependent processing of non-spliced epitopes in vitro, in vitro generation of spliced epitopes by PCPS may not exhibit the same fidelity and does not always simulate the efficacy of in vivo spliced epitope generation. This of course strongly questions the general application of the recently highlighted experimental pipeline for the identification of cancer-specific neo-splicetopes (*Mishto et al., 2019*). Reconsidering the workflow for the identification of neo-splicetopes, it thus seems that in vitro PCPS even when combined with peptide binding and TAP transport assays are not sufficient for the prediction of their immune relevance. We therefore believe that it is mandatory to first prove the cell surface presentation of algorithm-predicted candidate neo-splicetopes, either in humanized mice under conditions requiring processing and presentation or by peptide elution experiments, before TCR generation. Our data also support the notion (*Mylonas et al., 2018*) that the frequency of spliced epitopes is largely overestimated.

## Materials and methods

### Peptides, proteasome, and PCPS

The polypeptides substrates KRAS$^{G12V}_{2-14}$ (BIH 107) (TEYKLVVVGAVGV), KRAS$^{G12V}_{2-21}$ (Kloe 1178) (TEYKLVVVGAVGVGKSALTI), KRAS$^{G12V}_{2-32}$(TEYKLVVVGAVGVGKSALTIQLIQNHFVDEY), KRAS$^{G12V}_{2-35}$(TEYKLVVVGAVGVGKSALTIQLIQNHFVDEYDPT), as well as the RAC2$^{P29L}_{20-44}$ (BIH 5) polypeptide substrate (LISYTTNAFLGEYIPTVFDNYSANV) were synthesized by the core facility of the Institute of

Biochemistry (Dr. Petra Henklein) using standard Fmoc (*N*-(9-fluorenyl) methoxycarbonyl) methodology (0.1 mmol) on an Applied Biosystems 433A automated synthesizer. The peptide was purified by HPLC and analyzed by mass spectrometry (ABI Voyager DE PRO). The $KRAS^{G12V}_{1-21}$ (MDC 27) (MTE YKLVVVGAVGVGKSALTI) polypeptide substrate was obtained from JPT Peptide Technologies (Berlin, Germany). 20S proteasomes were purified from human red blood cells, LcL or T2.7 cells in principle following the procedure as previously described (*Textoris-Taube et al., 2019*). Proteasome composition of LcL and T2.7 cells, which express immunoproteasomes, however, may vary dependent on batch and culture conditions. For kinetic experiments and better comparison, therefore only the results obtained with erythrocyte 20S proteasomes were used for the kinetic experiments. Proteasome digests of the synthetic $RAC2^{P29L}$ and $KRAS^{G12V}$ polypeptides were performed in 100 µl of TEAD buffer (20 mM Tris, 1 mM EDTA, 1 mM $NaN_3$, 1 mM DTT, pH 7.2) over time at 37°C. For establishing a cleavage map for $RAC2^{P29L}_{20-44}$, processing times were 24 hr and 48 hr. The $RAC2^{P29L}_{20-44}$ and $KRAS^{G12V}_{1-21}$ synthetic polypeptide at a concentration of 60 µM was digested by 8 µg 20S proteasome. Proteasomal processing of the synthetic $KRAS^{G12V}_{2-21}$ and $KRAS^{G12V}_{2-14}$ polypeptides was performed at a substrate concentration of 40 µM or 60 µM and in the presence of 4 µg or 8 µg 20S proteasome, respectively. 10 µl digested sample was loaded for 5 min onto a trap column (PepMap C18, 5 mm × 300 µm × 5 µm, 100 Å, Thermo Fisher Scientific) with 2:98 (v/v) acetonitrile/water containing 0.1% (v/v) trifluoroacetic acid (TFA) at a flow rate of 20 µl/min and analyzed by nanoscale LC-MS/MS using an Ultimate 3000 and LTQ Orbitrap XL mass spectrometer (Thermo Fisher Scientific). The system comprises a 75 µm i.d. ×250 mm nano LC column (Acclaim PepMap C18, 2 µm; 100 Å; Thermo Fisher Scientific) or a 200 mm PicoFrit analytical column (PepMap C18, 3 µm, 100 Å, 75 µm; New Objective). The mobile phase (A) is 0.1% (v/v) formic acid in water and (B) is 80:20 (v/v) acetonitrile/water containing 0.1% (v/v) formic acid. For elution, a gradient 3–45% B in 85 min with a flow rate of 300 nl/min was used. Full MS spectra (*m/z* 300–1800) were acquired on an Orbitrap instrument at a resolution of 60,000 (FWHM). At first, the most abundant precursor ion was selected for either data-dependent collision-induced dissociation (CID) fragmentation with parent list ($1^+$, $2^+$ charge state included). Fragment ions were detected in an Ion Trap instrument. Dynamic exclusion was enabled with a repeat count of 2 and 60 s exclusion duration. Additionally, the theoretically calculated precursor ions of the expected spliced peptides were preelected for two Orbitrap CID (resolution 7500) and higher energy collisional dissociation (HCD) (resolution 15,000) fragmentation scans. The maximum ion accumulation time for MS scans was set to 200 ms and for MS/MS scans to 500 ms. Background ions at *m/z* 371.1000 and 445.1200 act as lock mass.

For LC-MS/MS runs using a Q Exactive Plus mass spectrometer coupled with an Ultimate 3000 RSLCnano (Thermo Fisher Scientific), samples were trapped as described above and then analyzed by the system that comprised a 250 mm nano LC column (Acclaim PepMap C18, 2 µm; 100 Å; 75 µm Thermo Fisher Scientific). A gradient of 3–40% B (alternatively 3–45% B) in 85 min was used for elution. The mobile phase (A) was 0.1% (v/v) formic acid in water and (B) 80% acetonitrile in water containing 0.1% (v/v) formic acid. The Q Exactive Plus instrument was operated in the data-dependent mode to automatically switch between full-scan MS and MS/MS acquisition. Full MS spectra (*m/z* 200–2000) were acquired at a resolution of 70,000 (FWHM) followed by HCD MS/MS fragmentation of the top 10 precursor ions (resolution 17,500, $1^+$, $2^+$, $3^+$, charge state included, isolation window of 1.6 *m/z*, normalized collision energy of 27%). The ion injection time for MS scans was set to maximum 50 ms, automatic gain control (AGCs) target value of $1 \times 10^6$ ions and for MS/MS scans to 100 ms, AGCs $5 \times 10^4$, dynamic exclusion was set to 20 s. Background ions at *m/z* 391.2843 and 445.1200 act as lock mass.

Peptides were identified by PD2.1 software (Thermo Fisher Scientific) based on their merged tandem mass spectra (MS/MS) of CID and HCD. For peptide identification, we set mass tolerances of either 10 ppm (for XL mass spectrometer) or 6 ppm (for Q Exactive mass spectrometer) on precursor masses and either 0.6 Da for fragment ions using Ion Trap or 0.06 Da using Orbitrap for fragmentation (for XL mass spectrometer) or 0.02 Da (for Q Exactive mass spectrometer).

In addition, for spliced peptides we compared the retention time and the merged MS/MS of CID and HCD with the fragmentation pattern of their synthetic counterparts. To identify spliced peptides, a fasta data file was generated with ProtAG for the $KRAS^{G12V}$ and $RAC2^{P29L}$ polypeptide substrates and loaded onto PD2.1. The kinetics were analyzed with LC Quan 2.7. HLA-A*02:01 binding

affinity of putative spliced epitopes was calculated by the netMHCpan 4.0 algorithm (*Jurtz et al., 2017*).

## Cell lines

T2 cells (ATCC: CRL-1992) were kept in RPMI supplemented with 10% fetal calf serum (FCS). The viral producer cell line HEK-GALV (HEK-293 cells stably expressing GALV-env and MLV-gag/pol) was cultured in DMEM supplemented with 10% FCS. Human peripheral blood mononuclear cells (hPBMCs) and Epstein–Barr virus–transformed lymphoblastoid B cell lines (B-LCLs; *Obenaus et al., 2015*) were cultured in RPMI 1640 supplemented with 10% FCS, 50 µM 2-mercaptoethanol, 1 mM sodium pyruvate, and non-essential amino acids. The human cell lines with KRAS$^{G12V}$ mutation were obtained from ATTC (AsPC-1 [ATCC: CRL-1682]; Capan-1 [ATCC: HTB-79]; CFPAC-1 [ATCC: CRL-1918]; NCI-H441 [ATCC: HTB-174]; NCI-H727 [ATCC: CRL-5815]; Panc 03.27 [ATCC: CRL-2549]; SW480 [ATCC: CRL-228]; SW620 [ATCC: CRL-227]) or Sigma-Aldrich (Colo668). The human cell lines carrying wildtype KRAS gene were MCF-7 (Sigma-Aldrich) and 624-Mel (RRID:CVCL_8054). K562 cells expressing HLA-A02:01, HLA-C07:01, and HLA-C07:02 molecules were obtained after transduction of K562 cells with retroviral vectors MP71 encoding the respective HLA molecules. The human melanoma cell line UKRV-Mel-21a (referred to hereafter as Mel21a) and the mouse cell line NIH-HHD have been described in *Sun et al., 2002* and *Popovic et al., 2011*, respectively. Cancer cell lines were kept in RPMI (Gibco) supplemented with 10% FCS (Pan Biotech), 1 mM L-glutamine, 1 mM sodium pyruvate, and non-essential amino acids. For HLA-A02:01 negative cell lines AsPC-1, Capan-1, NCI-H727, and Colo668 HLA-A02:01 expression was achieved by transient transfection with plasmid pMP71-A2 (retroviral vector encoding HLA-A*0201). All human tumor cell lines were authenticated by sequencing for the presence of the mutant or wildtype KRAS$^{G12V}$ configuration (*Chang et al., 2013*), and all experiments were performed with mycoplasma-free cells.

## Generation of neo-splicetope-specific T cells in ABabDII mice

For immunization (priming and successive boosts), ABabDII mice were injected subcutaneously with 100 µg of peptide (KLVVVGA<u>V</u>GV [KRAS$^{G12V}$-lin], KLVVVGA<u>V</u>GV [KRAS$^{G12V}$-sp1], KLVVVA<u>V</u>GV [KRAS$^{G12V}$-sp2], YLVVVGA<u>V</u>GV [KRAS$^{G12V}$-sp3], KLVVVG<u>V</u>GV [KRAS$^{G12V}$-sp4], F<u>L</u>GEYIPVF [RAC2$^{P29L}$ spliced], JPT) in a 200 µl 1:1 solution of incomplete Freund's adjuvant and PBS supplemented with 50 µg CpG. Repetitive immunizations were performed with the same mixture at least 3 weeks apart. KRAS/RAC2-specific CD8$^+$ T cells in the peripheral blood of immunized animals were assessed by in vitro peptide restimulation and subsequent intracellular cytokine staining (IFNγ) 7 days after each boost.

## Isolation and cloning of KRAS$^{G12V}$-sp1, KRAS$^{G12V}$-sp2-specific, and RAC2$^{P29L}$ splice-specific TCRs

Splenocytes and lymphocytes from inguinal lymph nodes were prepared from responding animals at day 8 after the last boost. For in vitro culture, CD4$^+$ T cells were depleted by CD4 microbeads (Miltenyi Biotech, Bergisch Gladbach, Germany) and $1 \times 10^6$ splenocytes were seeded per well of a 24-well plate and expanded for 10 days in RPMI 1640 medium supplemented with 10% FBS gold, HEPES, NEAA, sodium pyruvate, 50 µM β-mercaptoethanol, 20 IU/ml human IL-2, and $10^{-8}$ M sp1, sp2 or RAC2 peptide, respectively. Splenocytes were stimulated with $10^{-6}$ M peptide for 4 hr before mouse IFNγ secretion assay (Miltenyi Biotech). The cells were treated with Fc Block, stained with antibodies against mouse CD3-APC and mouse CD8-PerPC (BD Biosciences, San Jose, CA, USA). IFNγ secreting CD8$^+$ T cells were sorted with BD FACS Aria III (BD Biosciences) into RTL lysis buffer for RNA isolation with RNeasy Micro Kit (Qiagen, Hilden, Germany) according to the manufacturer's instructions. First-strand cDNA synthesis and 5'-RACE PCR were carried out using SMARTer RACE cDNA Amplification Kit (Clontech Laboratories) according to the manufacturer's instructions. In particular, subsequent TCR-specific amplification was carried out with 1–2 µl of the reverse transcription reaction, 1 U Phusion HotStart II polymerase (Thermo Scientific), 0.1 µM of either hTRAC (5'-cggccactttcaggaggaggattcggaac-3') or hTRBC (5'-ccgtagaactggacttgacagcggaagtgg-3') primers and 0.1 µM 5' primer (5'-ctaatacgactcactatagggcaagcagtggtatcaacgcagagt-3'). The amplified TCR genes were analyzed on an agarose gel and specific bands were cut out and cloned using a Zero Blunt TOPO PCR cloning kit (Life Technologies). Plasmids from individual clones were isolated and

sequenced using a T3 primer (5′–3′) at Eurofins Genomics. Dominant TCR-α/β chains were selected and paired as follows: 1376 TCR (TRAV5*01 – CAESTDSWGKLQF – TRAJ24*02, TRBV4-1*01 – CASSQDLAGYEQYF – TRBJ2-7*01), 9383B2 TCR (TRAV17*01 – CATDEDTGNQFYF – TRAJ49*01, TRBV12-3*01 – CASSLWGYEQYF – TRBD1*01 – TRBJ2-7*01), 9383B14 TCR (TRAV17*01 – CAT-DEDTGNQFYF – TRAJ49*01, TRBV12-3*01 – CASSLVGYEQYF – TRBD1*01 – TRBJ2-7*01), and 20967A2 TCR (TRAV20*02 – CAVQAPDSGNTPLVF – TRAJ29*01, TRBV2*01 – CASSDRGAYNEQFF – TRBD1*01 – TRBJ2-1*01). The TCR constant regions were replaced with mouse counterparts. Paired TCR-α/β chains were linked with a P2A element. TCR cassette was codon optimized, synthesized by GeneArt (Thermo Fisher Scientific, Waltham, MA, USA) and cloned into pMP71 by NotI/EcoRI restriction site cloning.

## TCR gene transfer

TCR gene transfer was carried out as described before (*Niedermann et al., 1995*). In brief, packaging cell line HEK-293-GALV (amphotropic) or Plat-E (ecotropic) were grown to approximately 80% confluence and transfected with pMP71 vector carrying the TCR cassette using Lipofectamine2000 (Life Technologies), and retrovirus-containing supernatant was harvested 48 hr and 72 hr after transfection.

Human PBMCs were isolated from healthy donors by Ficoll gradient centrifugation. $1 \times 10^6$ freshly isolated or frozen hPBMCs were stimulated with 5 µg/ml anti-CD3 (OKT3) and 1 µg/ml anti-CD28 (CD28.2) (BioLegend)-coated plates in the presence of 300 U/ml recombinant human interleukin 2 (hIL-2, Peprotech). Transductions were performed 48 hr and 72 hr after stimulation by addition of retrovirus-containing supernatant and 4 µg/ml protamine sulfate followed by spinoculation. Transduced T cells were kept in the presence of 300 U/ml hIL-2 for a total of 2 weeks followed by at least 2 days of culture in the presence of 30 U/ml hIL-2, before they were used for experiments.

For transduction of mouse T cells, spleen cells were isolated from TCR1xCD45.1xRag1$^{-/-}$ mice (expressing a monoclonal-irrelevant TCR against SV40 large T), erythrocytes were lysed, and cells were stimulated in medium (RPMI 1640, 10% FCS, 100 IU/ml penicillin-streptomycin, 1 mM sodium pyruvate, 1× non-essential amino acids, 50 µM 2-mercaptoethanol) supplemented with 1 µg/ml anti-mouse CD3, 0.1 µg/ml anti-mouse CD28 antibodies (both BD Biosciences (BD), Franklin Lakes, NJ, USA), and 10 IU/ml human IL-2 (Proleukin S, Novartis, Basel, Switzerland) at a concentration of $2 \times 10^6$/ml. $1 \times 10^6$ cells were transduced twice by spinoculation in the presence of 10 IU/ml IL-2 and $4 \times 10^5$ mouse T-Activator beads (Life Technologies). T cells were expanded in medium (+50 ng/ml IL-15; Miltenyi Biotec) for 10 days before co-culture.

## Functional assays

IFNγ production was measured by ELISA after 16 hr co-culture of $1 \times 10^4$ TCR-positive T cells with $1 \times 10^4$ target cells (human/mouse tumor cell lines or peptide-loaded T2 cells). Stimulation with phorbol myristate acetate (PMA) and ionomycin was used as a positive control. All samples were measured in duplicate.

## Flow cytometry

The following conjugated antibodies were obtained from BioLegend: anti-hCD3 (HIT3α), anti-hCD8 (HIT8α), anti-hHLA-A2 (BB7.2), anti-hHLA-ABC (W6/32), anti-mCD3 (145-2 C11), anti-mCD8 (53-6.7), anti-mIFNγ (XMG1.2), and anti-mTCR-β (H57-597). Samples were analyzed using MACSQuant (Miltenyi) or FACSCalibur (BD Biosciences). Data analysis was performed using FlowJo (Treestar).

## Quantitative PCR (qPCR)

RNA was isolated from $1 \times 10^6$ NIH-HHD cells using the NucleoSpin TriPrep (MACHEREY-NAGEL). After RNA quality and integrity were verified, 2.8 µg of total RNA were used as template for cDNA synthesis with random hexamers, using SuperScript III Reverse Transcriptase (Invitrogen). Samples were diluted 1:5 and 5 µl used as template in a 20 µl qPCR reaction using SYBR Green PCR Master Mix (Thermo), with 500 nM primer concentration. Primer sequences: GFP: F:5′-acgacggcaactacaagacc-3′, R:5′-tgaagtcgatgcccttcag-3′; *Gapdh*: F:5′-tggagaaacctgccaagtatg-3′, R:5′-gttgaagtcgcaggagacaac-3′. Samples were run on the QuantStudio 3 Real-Time PCR system (Thermo Fisher) and analyzed according to the comparative ΔΔCt method.

## ProtAG prediction algorithm

The ProtAG prediction algorithm was used in combination with mass spectroscopy to identify peptides and spliced peptides derived from an oligomeric protein substrate. The peaks of the MS intensity profile were approximated by Gaussian functions. Goodness of fit was used as one criterion for assessing the reliability of a mass peak. Only peaks above a user-defined noise threshold were compiled together with their HPLC retention times. The likelihood for correctly assigning a peptide to an MS peak was scored by the correspondence between computational and experimental values for peptide mass, occurrence of expected $m/z$ values, similarity of retention times, and tandem MS/MS data. Chemically modified peptides (e.g., by oxidation) were identified by adding to the theoretical mass the masses of possible modifiers. Such modified peptides were included into the list of identified peptides only if the non-modified peptide could also be identified. After assigning the MS peaks to all direct or chemically modified fragments that theoretically can be derived from the protein substrate by one or multiple cleavages, a group of significant but 'unexplained' MS peaks remains, which may represent possible spliced products, that is, peptides composed of fragments distant in the parental protein substrate. The likelihood for correctly assigning a splice peptide to unexplained MS peaks was computed in the same way as for conventional peptides, including the additional criterion, that the two fragments merged together in the presumed splice peptide were also present in the set of identified conventional peptides.

Proteasomal cleavage products (PCP) of a substrate peptide can clearly be described by the numbers of the first and last amino acid within the substrate: P(i,j) denominates the peptide of length j-i +1 starting with amino acid i and ending with amino acid j. Proteasomal splice products (PSP) consist of two such fragments, therefore denominated by SP(i,j,k,l), consisting of the peptides P(i,j) and P(k,l) and having a length of (j + l-i-k + 2). They can be in normal order (i < j < k < l), inverse order (k < l < i < j), or overlapping. Overlapping splice product means that there exists a position m that is both part of P(i,j) and P(k,l), therefore max(i,j) <= min(j,l), meaning that the splice product consists of parts of two substrate peptides.

Splice peptides in normal order with j + 1 = k are identical to the original PCP P(i,l) and should therefore be excluded. Splice peptides can have the same sequence like PCPs, for example, if the sequence of the substrate is redundant, or the length of one of the parts is short – such peptides can be excluded from the database. Splice peptides can have the same sequence, so the splice peptides SP(i,j,k,l) and SP(i,j + 1,k + 1,l) have the same sequence if the amino acid in position j + 1 and in position k are the same. Nevertheless, both versions should be kept within the database because if the original peptides P(i,j) and P(k,l) are found within the cleavage products, and P(i,j + 1) not, the first version of the PSP is more likely.

When searching for splice products that are at the same time epitopes for MHC class I or MHC class II, the length of the predicted splice products should be limited. Therefore, according limits are included into the algorithm. Also, the database of spliced products can become very large if you try to evaluate all possible splice products of a large substrate without limits. The number of all possible splice products of a substrate of length 100 consists of about 25 million peptides, and results into a database of nearly 3 GB, so you can predict the size of the database without evaluating it, and avoid the evaluation.

The ProtAG program evaluates a database of splice products in fasta format according different parameters:

- Sequence of the substrate of length L(sub).
- Minimal/maximal length of the parts of the splice peptides (MinP,MaxP).
- Minimal/maximal length of the gap between the parts of the splice peptide (only used if you evaluate splice peptides in normal or inverse order) (MinG,MaxG).
- Minimal/maximal length of the whole splice peptide (MinS,MaxS).
- Do you want to include PCPs into the database (recommended)?
- Do you want to exclude PSPs with sequences identical to PCPs (recommended)?
- Do you want to evaluate only PSPs in normal order, or PSPs in normal or inverse order (coming from the same substrate), or all PSPs including PSPs from different substrates?

## Algorithm

Algorithm for the evaluation of PSPs in normal or inverse order – ignoring MaxG for simplification:

– evaluate all possible lengths of splice products actL with MinS<=actL<=MaxS

– evaluate possible lengths for the parts: MinP<=actL1<=min(actL-MinP,MaxP), actL2=actL-actL1

– evaluate starting points for the first fragment of the splice peptide: 1<=Start1<=L(sub)-actL-MinG+1

endpoint for the first fragment will be End1=Start1+actL1-1

– evaluate starting points for the second fragment: End1+1+MinG<=Start2<=L(sub)-actL2+1

endpoint for the second fragment will be End2=Start2+actL2-1

– evaluate the normal PSP SP(Start1,End1,Start2,End2) if Start2 >End1+1

check if the sequence of the PSP is a sub sequence of the substrate

– if not: write out the PSP in fasta format with the name line containing information to the positions

Start1, End1, Start2, End2, mass, m/z values for z=1,2,3

– if inverse PSPs should be included: evaluate the inverse PSP SP (Start2, End2, Start1, End1)

– check if the sequence of the PSP is a sub sequence of the substrate

if not: write out the PSP in fasta file format

The ProtAG algorithm, together with an instruction sheet is available on Dryad.

## Acknowledgements

We thank Sabrina Horn, Kathrin Borgwald, and Mathias Pippow for excellent technical support. We acknowledge the participation of Michele Mishto (postdoc in CRG-1) and Juliane Liepe (Imperial College, London) in the initial splicing experiments. This work was supported by grants from the German Research Foundation (SFB-TR36), the Berlin Institute of Health (CRG-1), Einstein Stiftung (A-2013-174), Berliner Krebsgesellschaft, Deutsche Krebshilfe (111546), DKTK joint funding (NEO-ATT), and the Helmholtz-Gemeinschaft, Zukunftsthema 'Immunology and Inflammation' (ZT-0027).

## Additional information

### Funding

| Funder | Grant reference number | Author |
| --- | --- | --- |
| German Research Foundation | SFB-TR36 | Gerald Willimsky Thomas Blankenstein |
| Deutsche Krebshilfe | 111546 | Gerald Willimsky |
| Einstein Stiftung Berlin | A-2013-174 | Peter M Kloetzel |
| German Cancer Research Center | NEO-ATT | Gerald Willimsky |
| Berliner Krebsgesellschaft | | Peter M Kloetzel |
| Helmholtz Association | ZT-0027 | Thomas Blankenstein Gerald Willimsky |
| Berlin Institute of Health | CRG-1 | Thomas Blankenstein Peter M Kloetzel |

The funders had no role in study design, data collection and interpretation, or the decision to submit the work for publication.

### Author contributions

Gerald Willimsky, Conceptualization, Data curation, Formal analysis, Supervision, Funding acquisition, Investigation, Methodology, Writing - original draft, Project administration; Christin Beier, Lena Immisch, George Papafotiou, Vivian Scheuplein, Investigation, Methodology; Andrean Goede,

Hermann-Georg Holzhütter, Software, Methodology; Thomas Blankenstein, Peter M Kloetzel, Conceptualization, Funding acquisition, Writing - original draft

### Author ORCIDs
Gerald Willimsky https://orcid.org/0000-0002-9693-948X
Andrean Goede https://orcid.org/0000-0002-9044-9869

### Ethics
Animal experimentation: All animal experiments were performed according to institutional and national guidelines and regulations. The experiments were approved by the governmental authority (Landesamt für Gesundheit und Soziales, Berlin, H0086/16).

### Decision letter and Author response
Decision letter https://doi.org/10.7554/eLife.62019.sa1
Author response https://doi.org/10.7554/eLife.62019.sa2

## Additional files

### Supplementary files
• Supplementary file 1. HLA-ABC haplotypes of lymphoblastoid B cell lines (BLCLs) and tumor cell lines SW480 and SW620.
• Supplementary file 2. Faulty peptides identified within the KRAS$^{G12V}$ polypeptide substrates.
• Transparent reporting form

### Data availability
Additional source data comprising databases for ProteomDiscoverer, KRAS/RAC2 kinetics, cleavage maps and PD2.1 result files have been submitted to Dryad under https://doi.org/10.5061/dryad.jq2bvq88b. The ProtAG algorithm and an instruction sheet are also available on Dryad.

The following dataset was generated:

| Author(s) | Year | Dataset title | Dataset URL | Database and Identifier |
|---|---|---|---|---|
| Willimsky G, Beier C, Immisch L, Scheuplein V, Goede A, Holzhütter HG, Blankenstein T, Kloetzel PM | 2021 | Willimsky_et_al_11-08-2020-RA-eLife-62019_Supplementary data files | http://dx.doi.org/10.5061/dryad.jq2bvq88b | Dryad Digital Repository, 10.5061/dryad.jq2bvq88b |

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
