## [Decision Letter]

Thank you for submitting your article "in vitro proteasome processing of neo-splicetopes does not predict their presentation in vivo" for consideration by *eLife*. Your article has been reviewed by 2 peer reviewers, and the evaluation has been overseen by a Reviewing Editor and Tadatsugu Taniguchi as the Senior Editor. The following individual involved in review of your submission has agreed to reveal their identity: Michal C Bassani (Reviewer #1).

The reviewers have discussed the reviews with one another and the Reviewing Editor has drafted this decision to help you prepare a revised submission.

Summary:

The manuscript by Willimsky et al. describes an intensive experimental assessment of the (lack of) immunogenicity of predicted neo-splicetopes covering two common driver mutations. Following prediction of potential proteasomal spliced peptides covering two mutations, the authors identified by MS spliced-products in in-vitro digest assays, and they generated specific TCRs against these epitopes in in-vivo mice models. With these TCR, they concluded lack of natural presentation of the spliced peptides in vivo, also in cells overexpressing the mutation.

Globally the study is well performed and demonstrates very convincingly that algorithms combined with in vitro precursor peptide digestions are by no means a guarantee that spliced epitopes will be generated in cellulo and in vivo. The results concerning the mutant KRAS epitope contradict a recently published report claiming, in the discussion and without showing data, that the spliced KRAS(G12V) epitope is presented by tumor cells (Mishto 2019 in the reference list).

The dilemma with this report is of course that it shows "negative data". However, it appears that the report will be of interest to a broad public. There are numerous examples in the literature where T cell epitopes are lightheartedly taken for real without hard evidence as pursued in this report. Moreover, the issue of the in vivo relevance of proteasome-produced spliced epitopes remains controversial since several years and this report adds a cautionary note to this ongoing discussion.

All in all, although this is an n=2 study, this work adds important experimental validation to an ongoing debate about the fidelity, the prevalence and the clinical importance of proteasomal spliced peptides. Overall, the data reported is convincing that there is no natural presentation of these selected neo-splicetopes. However, it is important to note that the readers might be left with a feeling that these were just two unlucky examples.

Essential revisions:

The authors should address the following points raised by the reviewers:

1. Lines 129-140. The information reported in this section is highly important to understand the validity of the previous paper (Mishto et al. 2019). However, it is unclear how the authors obtained the information that there was a faulty peptide synthesis and technical issues with the in-vitro splicing experiments in that paper ("in the initial PCPS experiments"). The authors should provide more clarifications and data to support these arguments of "contaminating peptides mimicking a PCPS reaction". A discussion should be added on how prevalence are these contaminating peptides, and what are the quality control assays that should be included to exclude such critical technical issues in future experiments.

2. Lines 143-151 and 267-281: Here the authors report that the spliced peptides were identified by MS in the in-vitro assays. These section should be extended and supported with more data to report how efficient were these splicing reaction. For example – how many copies of each of the spliced peptides were obtained compared to the non-spliced products? Annotated MSMS should be provided for all the spliced peptides. For example, in Figure 1B, there are many dominant peaks that were not annotated, and it is impossible for the reader/reviewer to assess the quality of this identification, which overall looks not great. Each of the identified peptides (spliced and non-spliced) should be reported in a table, with quality scores that are generated by the database search tool (in this case the PD2.1 tool), such as accuracy, intensity in each of the time points in the kinetics assays, identification score, FDR, score to second bet match etc. MS raw data should also be deposited and available for the readers to explore. Standard parameters about the database search should be provided, such as the reference database used (the spliced targets and which baseline reference?), FDR, mass deviation for parent mass and for precursor ions, etc.

3. While the authors make a very convincing case that tumor cells do not present the two epitopes, they might consider corroborating the conclusion that even over-expression of the source antigen does not allow for epitope presentation, i.e. that lack of presentation is not due to limiting intracellular epitope amounts. Presently this conclusion is based on the fact that transfection of the 35mer also fails to induce presentation (Figure 4). However, this construct encodes an artificial tandem peptide and it is unclear whether digestion of this peptide would yield the spliced epitope in vitro. In other words, without an in vitro digestion of this peptide, the argument that in vitro digestion is unsuitable for predicting processing in cellulo cannot be made, for the situation of over-expressed source antigens.

4. Moreover, the authors do not make any effort to estimate the amount of the 35mer expressed; directly quantifying expression levels would make their argument more solid.

---

## [Author Response]

Essential revisions:The authors should address the following points raised by the reviewers:1. Lines 129-140. The information reported in this section is highly important to understand the validity of the previous paper (Mishto et al. 2019). However, it is unclear how the authors obtained the information that there was a faulty peptide synthesis and technical issues with the in-vitro splicing experiments in that paper ("in the initial PCPS experiments"). The authors should provide more clarifications and data to support these arguments of "contaminating peptides mimicking a PCPS reaction". A discussion should be added on how prevalence are these contaminating peptides, and what are the quality control assays that should be included to exclude such critical technical issues in future experiments.

We have rewritten this part of the manuscript giving more detailed and extended information on how we obtained the information that there was faulty peptide synthesis and we also included additional experimental data. In this context it is important to note that the KRAS^G12V^ polypeptide substrates in question, i.e. KRAS^G12V^_2-32,_KRAS^G12V^_2-21_ and KRAS^G12V^_2-14_ (newly added) as well as KRAS^G12V^_2-35_ (the substrate used by Mishto et al.) were synthesized by the same Charité core facility.

Initially we were misled because generation of the KRAS^G12V^ derived spliced epitopes in our PCPS experiments perfectly followed the kinetics expected for the generation of spliced peptides (shown in Figure 3—figure supplements 4 and 5A). Early concerns regarding the fidelity of our experiments were raised by the negative in vivo experiments. Also, when we routinely searched the MS data from our kinetic PCPS experiments for N-terminally or C-terminally elongated precursors of the spliced epitopes generated during the PCPS reaction, we detected several long polypeptides that already in the absence of any processing event reflected the sequence of the predicted spliced peptides (now summarized in Supplementary File 2). This strongly indicated that these faulty polypeptides could have served as substrates for a normal proteasomal processing reaction resulting in the production of non-spliced peptides mimicking a splicing reaction.

Prevalence

As can be seen in Figure 3—figure supplements 5B, C these faulty KRAS^G12V^ derived polypeptides, KRAS ^G12V^_2-8_10-21_ and KRAS ^G12V^_2-10_12-21_ are degraded by the proteasome with kinetics similar to that of the total polypeptide substrate reflecting cleavage and generation of linear non-spliced peptides.

Considering, that in vitro generation of non-spliced epitopes is in most cases significantly more efficient than the generation of spliced epitopes, i.e. for RAC2^P29L^ approx. 200-fold (numbers in the literature vary between 100- to 10000-fold) even minor amounts of false substrate peptide may lead to the generation of prevalent false spliced peptides. As requested this aspect is now included and discussed in the manuscript.

2. Lines 143-151 and 267-281: Here the authors report that the spliced peptides were identified by MS in the in-vitro assays. These section should be extended and supported with more data to report how efficient were these splicing reaction. For example – how many copies of each of the spliced peptides were obtained compared to the non-spliced products? Annotated MSMS should be provided for all the spliced peptides. For example, in Figure 1B, there are many dominant peaks that were not annotated, and it is impossible for the reader/reviewer to assess the quality of this identification, which overall looks not great. Each of the identified peptides (spliced and non-spliced) should be reported in a table, with quality scores that are generated by the database search tool (in this case the PD2.1 tool), such as accuracy, intensity in each of the time points in the kinetics assays, identification score, FDR, score to second bet match etc. MS raw data should also be deposited and available for the readers to explore. Standard parameters about the database search should be provided, such as the reference database used (the spliced targets and which baseline reference?), FDR, mass deviation for parent mass and for precursor ions, etc.

Because our intention was to focus more on the in vivo data the part concerning peptide splicing was kept rather short. But we fully agree with the reviewer that the lack of documentation left many questions open. As requested, we now deposited MS raw data, PD2.1 files and quality scores used for the identification of spliced and non-spliced peptides in our KRAS^G12V^ and RAC2^P29L^ kinetic PCPS experiments as well as cleavage map for the RAC2^P29L^ substrate.

Since we also extended the number of documented experiments considerably, we adapted the material and methods section accordingly.

For example, in Figure 1B, there are many dominant peaks that were not annotated, and it is impossible for the reader/reviewer to assess the quality of this identification, which overall looks not great.

The reviewer is absolutely correct as this made us aware of a critical embarrassing mistake which changed the conclusion of our KRAS^G12V^ experiments and for which we have to deeply apologize. It shouldn´t have happened. Thorough reanalysis of the MSMS data generated in the kinetic experiments performed with the new KRAS^G12V^_1-21_ polypeptide substrate made us realize that in our database two peptides with identical masses were proposed. Thus, the MS/MS spectrum shown in the Figure 1B in the initial manuscript is the spectrum for KRAS^G12V^ ORI_5-13 KLVVVGAVG and not as falsely assumed KRAS^G12V^ 58_10-14 KLVV_GAVGV. The dominant peaks at 383.45 and 539.11 can clearly only be identified in KRAS^G12V^ ORI_5-13 KLVVVGAVG and not in KRAS^G12V^ 5-8_10-14 KLVV_GAVGV (synthetic peptide). Consequently and in line with our in vivo data we have thus no experimental evidence that the predicted KRAS^G12V^_5-8/10-14_ neo-splicetope is generated.

For example – how many copies of each of the spliced peptides were obtained compared to the non-spliced products?

The main interest of the project relied in the identification of potentially new spliced tumor epitopes at the qualitative level. Therefore, we didn´t perform any titration experiments with spiked peptides which would have allowed a reliable estimation of peptide number. As now pointed out in the manuscript the absolute numbers of peptides generated in vitro is strongly dependent on a number of different parameters such as substrate concentration, type of substrate, amount and activity of proteasomes as well as assay time. Importantly, the experimental conditions for a PCPS reaction by no means reflect the in vivo situation where only a single substrate molecule occupies the catalytic cavity at any given time. In contrast, in vitro peptide splicing reactions as a result of transpeptidation require relatively high substrate concentrations and are at large driven by a high substrate density in the cavity. This bears the risk of artificial splicing reactions as evidenced in by the formation of a reversed spliced peptide with sequence overlap RAC2^P29L^_28-31/24-28_ (Mai et al., 2020, cleavage map RAC2^P29L^). Therefore, and with respect to judging splicing efficiency for a specific spliced epitope, for example RAC2^P29L^_28-34/36-37_ (Figure 4A) it seems appropriate to compare the generation of a spliced epitope with that of the non-spliced epitope. For RAC2^P29L^ this means that even under conditions that favor peptide splicing the RAC2^P29L^ non-spliced neo-epitope is at least 200-fold more efficiently generated than RAC2^P29L^_28-34/36-37_, as judged by the comparison of ion currents. [We also calculated the molecule numbers generated in an independent 24h experiment using the Avogadro constant. In this case generation of the non-spliced epitope is approximately 1000 more efficient than the generation of the spliced epitope].

3. While the authors make a very convincing case that tumor cells do not present the two epitopes, they might consider corroborating the conclusion that even over-expression of the source antigen does not allow for epitope presentation, i.e. that lack of presentation is not due to limiting intracellular epitope amounts. Presently this conclusion is based on the fact that transfection of the 35mer also fails to induce presentation (Figure 4). However, this construct encodes an artificial tandem peptide and it is unclear whether digestion of this peptide would yield the spliced epitope in vitro. In other words, without an in vitro digestion of this peptide, the argument that in vitro digestion is unsuitable for predicting processing in cellulo cannot be made, for the situation of over-expressed source antigens.

An in vitro digest of repetitive sequences and its interpretation is highly questional. For analysis of T cell recognition of the neo-spicotope RAC2^P29L^ – that we also confirmed by in vitro proteasomal digest – we now include cell lines that overexpress RAC2^P29L^ cDNA excluding any artificial tandem peptide generation. Since the TCR 22894, specific for the non-splice epitope, comparatively recognizes RAC2^P29L^ triple 45mer (Figures 5A/C) as well as RAC2^P29L^ cDNA (Figures5B/C) we prove sufficient generation of the non-splice epitope from both recombinantly expressed expression cassettes. Splice-specific TCR 20967A2 exclusively only recognizes the overexpressed RAC2^P29L^ triple epitope nonamer (Figures 5A/C), supporting our notion that even if the RAC2^P29L^_28-34/36-37_ neo-splicetope is derived from an overexpressed substrate protein, its amounts are negligible and insufficient to allow its recognition by T cells.

4. Moreover, the authors do not make any effort to estimate the amount of the 35mer expressed; directly quantifying expression levels would make their argument more solid.

Using quantitative PCR we analysed expression of multimer peptide minigenes (KRAS^G12V^ 35mer, RAC2^P29L^ 45 mer) in comparison to our epitope minigenes (Figures 3E and 5E). Whereas the KRAS^G12V^ 35mer shows two-fold higher expression than the spliced KRAS^G12V^ 9mer, expression of the RAC2^P29L^ 45 mer is five-fold stronger than expression of the spliced RAC2^P29L^ 9mer. Therefore we conclude that – despite overexpression – still insufficient, if at all, amounts of a putative neo-splicotope are produced from the multimers to be recognized by respective T cells.